# A neurocognitive mechanism for increased cooperation during group formation
Wojciech Zajkowski [1,2,6] ✉, Ryan P. Badman[1,3,4,6], Masahiko Haruno[5] & Rei Akaishi [1] ✉

How do group size changes influence cooperation within groups? To examine this question, we performed a dynamic, network-based prisoner's dilemma experiment with fMRI. Across 83 human participants, we observed increased cooperation as group size increased. However, our computational modeling analysis of behavior and fMRI revealed that groups size itself did not increase cooperation. Rather, interaction between (1) participants' stable prosocial tendencies, and (2) dynamic reciprocal strategy weighed by memory confidence, underlies the group size-modulated increase in cooperation because the balance between them shifts towards the prosocial tendency with higher memory demands in larger groups. We found that memory confidence was encoded in fusiform gyrus and precuneus, whereas its integration with prosocial tendencies was reflected in the left DLPFC and dACC. Therefore, interaction between recall uncertainty during reciprocal interaction (i.e., forgetting) and one's individual prosocial preference is a core pillar of emergent cooperation in more naturalistic and dynamic group formation.

Cooperation within groups is the cornerstone of human civilization. Since the foundation of economic sciences, especially after the introduction of game theory[1], much work has been done to explain how inherently selfish agents may establish and sustain mutually beneficial cooperative relations[2–5]. However, social value-related studies typically focus on static groups where group membership is forced and stationary, which may fail to capture important real world group dynamics where humans can leave groups they dislike, sanction uncooperative group members, and observe group membership fluctuate. Additionally, agents in groups are usually assumed to either have perfect memory or little-to-no memory in these settings. In reality, humans have decent but noisy working memory ability[6], and are typically self-aware about the potential for error during recall of past events[7]. Thus, little is known about how the dynamics of group formation itself affect individual behavior, including how memory capacity modulates decision strategies and overall levels of cooperativeness in larger group sizes[8].

Group size, reflected in the number of close interpersonal relationships, has been associated with brain size and structure on evolutionary timescales. It correlates with brain size across primate species[9], while individual differences in white matter structure predict the number of friends an individual has[10]. Experimental manipulation in macaques has shown that controlling group size during development affects brain maturation, with larger groups resulting in expanded gray matter volume within regions associated with social functions[11]. Together, these findings provide convincing evidence in favor of group size affecting brain structure and behavior on the timescales of lifetimes and generations. However, the evolutionary perspective offers little insight into the underlying behavioral policies and brain circuits that compute whether or not to cooperate within groups during an organism's lifetime, especially in daily life timescales and decision-making events. Here, we investigate how group size influences cooperation on the scale of minutes to hours within a dynamic group laboratory task, in order to simulate the early process of group formation. Unlike the bulk of prior work, we focus on the associated dynamics of sequential pairwise (dyadic) cooperation between individual members within the group context, which is a prevalent type of social interaction[12]. Furthermore, these dyadic interactions take place in groups of up to six total participants (the participant plus up to five partners), a group size relevant for team sizes in research, engineering or business[13]. Additionally, we perform functional magnetic resonance imaging (fMRI) scanning during the task to investigate whether the dynamics of group growth induce functional brain changes and

[1]Social Value Decision-Making Collaboration Unit, RIKEN Centre for Brain Science BTCC TOYOTA Collaboration Center, Wako, Saitama, 351-0198, Japan. [2]Laboratory of Sensorimotor Research, National Eye Institute, National Institutes of Health, Bethesda, MD, USA. [3]Department of Neurobiology, Harvard Medical School, Boston, MA, 02115, USA. [4]Kempner Institute, Harvard University, Boston, MA, 02134, USA. [5]Center for Information and Neural Networks, National Institute of Information and Communications Technology, Suita, Osaka, 565-0871, Japan. [6]These authors contributed equally: Wojciech Zajkowski, Ryan P. Badman. ✉e-mail: wojciech.zajkowski@nih.gov; akaishirei@gmail.com

different behavioral responses that mirror or support the brain structure changes seen on evolutionary timescales.

From a normative standpoint, and supported by much prior game theory work in group paradigms, larger group sizes generally discourage cooperation[3,14]. Exploiting individual partners becomes more viable, since losing one cooperative partner has less opportunity costs the more remaining partners we have. Additionally, greater interaction distance (less frequent interactions with a particular partner) lessens the incentive to cooperate by decreasing the total potential reward from cooperating in the future[15]. Most experiments in this area focus on static designs where group size is externally manipulated and fixed between participants, and indeed report less cooperation in larger groups[16–18].

From an algorithmic perspective however[19], larger interaction distance (a natural consequence of larger groups) might affect cooperation by interfering with a reciprocity-based strategy. The most famous of such strategies being *Tit-for-Tat* (TFT), a simple but "fair" policy of deterministic reciprocation, which has proven to be extremely effective and robust in many settings[15], and is prevalent among human agents both because of its performance and due to inherent human valuation of fairness[20]. In order to reciprocate, one needs to remember their partner's previous choice. As the group gets larger, one has more pairwise interactions to remember, and each of them is further apart in time if the pairwise interactions are sequential (as often occurs in the real world) rather than simultaneous. This situation leads to more memory errors[21], making reciprocity increasingly difficult to implement. Previous work has shown that constraining memory capacity led participants to use a forgiving variant of TFT, *generous TFT* (gTFT), where partner's defections are met with a low-to-moderate probability of forgiveness[22]. This forgiving adjustment to TFT can prove effective in fostering a highly cooperative environment[4]. A probability of forgiveness also protects from the deadly effects of an reverberating echo[15], where previously cooperative partners fall out of synch and follow with either a cycle of alternating between defection and cooperation ignited by a single mistake, or alternatively a cycle of defection. Hence, if a reciprocating agent prioritizes the preservation of a mutually cooperative relationship while being uncertain about the partner's previous choice, he or she should become more forgiving. Furthermore, introducing some stochasticity to the forgiveness choices can help prevent uncooperative cycles that can result in mismatched pairwise deterministic strategies[23].

Computationally, such a strategy could be implemented in at least two ways. A simple heuristic approach requires keeping track of the group size and adjusting behavior according to the increased cognitive load. This way, one would counteract the poorer (on average) memory reliability in larger groups by cooperating more frequently as a safe default behavior, irrespective of interaction history, effectively becoming more prosocial. Alternatively, one might track the reliability directly. Such an agent would thus selectively alter their behavior by weighing the tendency to reciprocate by the reliability of the memory of their partner's previous choice. This approach is more computationally expensive, as it requires integrating timescale-sensitive information about previous partner-specific interactions (which could have occurred recently, or many trials ago). On the other hand, this approach is potentially more effective, as it allows the preservation of high levels of reciprocity when partner-related information is certain.

While dynamic groups (where group size and composition can be influenced by participants) were shown to facilitate cooperation[24,25], typically by allowing them to create static groups after early rejection of defective partners, no previous studies tested this effect within group-embedded dyadic interactions and consistently dynamic group membership that is more naturalistic, nor attempted to explain the neurocognitive mechanism driving behavioral change. Here, we tackle this problem on three levels. (1) To test the qualitative pattern of behavior, we utilize the unique network structure of the iterated prisoner's dilemma (PD) task (Fig. 1A, B). In each trial, a participant interacts with a randomly chosen group member. On randomly selected trials, a new partner is introduced at the start, or players are given the opportunity to unilaterally exclude the current partner from their group after the PD interaction results are shown. Introduction of these

agentic control mimics natural settings, where people are free to associate with chosen partners and avoid others, aiding in the formation of cooperative groups[26–28]. (2) To better understand the underlying cognitive processes, we formalize our hypotheses within a value-based decision-making computational framework, which describes how subjective values are formed and mapped to choices. We then test the hypothesized cognitive mechanisms by comparing model fits. (3) To understand how these processes are implemented, we map the model-based predictions to fMRI-derived brain activity. Together, these findings provide a neurocognitive mechanism for the emergence and maintenance of cooperation in a dynamic group setting.

## Methods
### Task design
**Overview**. Our network-embedded-dyad prisoner's dilemma (NEDPD) task uses a dynamic social network, with semi-frequent opportunities to break social links (at the end of ~20% of the trials), while a newcomer to the network is gained at the start of ~10% of the trials automatically (Fig. 1). Participants play dyadically with either the new group member on a newcomer trial, or with a randomly chosen group member on a non-newcomer trial. The number of the in-person participants per session group varied from 3 to 6. These task parameters have been selected in order to promote cooperative behavior[17,25,29] (Fig. S1, S2). Participants were told they played the PD game with the other participants in their session group, whose true identity was concealed with avatar faces (Fig. 1). Faces are taken from the neutral category of the *NimStim* set of facial expressions[30]. Unbeknownst to participants, for detailed control of the experimental conditions, they were actually engaged in the game with prescheduled computer programs (see the *Algorithm of Social Partner* section below). See *Effects of Partner Face* section of the *Supplementary Materials* for control analyses regarding effects of specific faces on behavior.

**Participants, payment, exclusion criteria**. A total of 87 (49 male; 38 female; self-reported biological sex) healthy volunteers were recruited in the behavioral experiments. Participants were required to provide informed consent before experiments. Our study was approved by the review board at the National Institute of Information and Communications Technology in Osaka, Japan. All participants had normal or corrected-to-normal vision and were screened for the presence of psychiatric or neurological disorders. All participants were native Japanese. Among them, 26 participants participated in the behavioral task while simultaneously undergoing functional magnetic resonance imaging (fMRI), while the other 61 participants participated in the behavioral experiment only outside of the fMRI. 4 participants (all participating in the behavioral-only session) were excluded from further analyses due to defecting on the vast majority (>95%) of trials. While we acknowledge that 'always defect' is a viable strategy used by a subset of population in prisoner's dilemma games[20], this a priori selection criterium served to remove participants who were completely unreceptive to the experimental manipulation. It is also very difficult to model participants who always make the same choice and would severely affect statistical sampling in the fMRI analysis to have little-to-no contrast in behavior. For 25/83 participants, their age was not recorded though all participants were visibly young adult students (~20–30 years old). The mean age of the 62 participants with recorded age information was 22.0 ± 1.8 (SD). All participants gave informed consent in accordance with the CiNet Research Ethics Committees. Participants were paid 3000 yen (approximately $30 US dollars during the time of the experiment) for their participation in the behavioral experiments and 5000 yen (approximately $50 US dollars) for the participation in fMRI experiments. The study was not preregistered.

**Session start conditions**. The sessions lasted for 180 trials, and the average completion time was around 45 min. Before each session, the

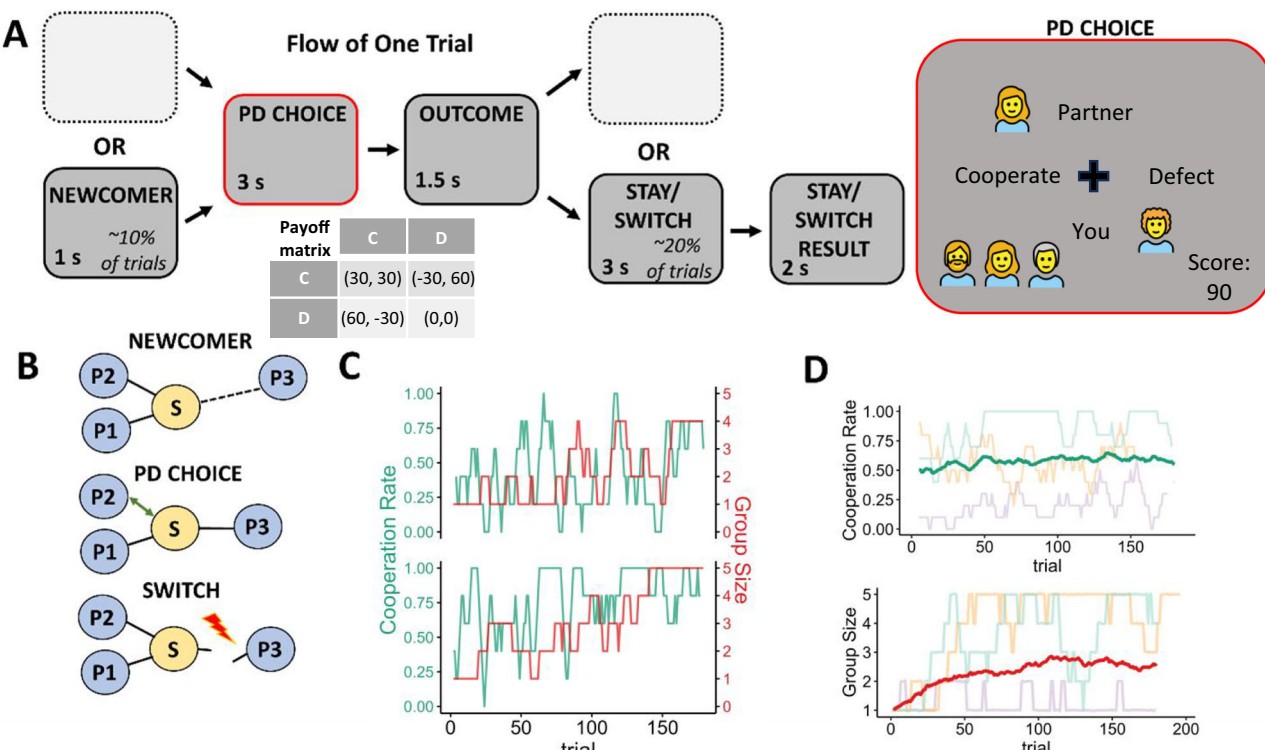

**Fig. 1 | Task design. A** Steps of a single trial in the embedded-dyad network Prisoner's Dilemma (NEDPD) task, with (right insert) a recreation of the PD choice task screen shown that contains current group members' avatars, the subject's avatar, and the cumulative score. The experimental avatars consisted of photos from the *NimStim* database[30]. Here we use images from the open access OpenMoji (openmoji.org) database as their symbolic representation. **B** A cartoon schematic of task components described in (**A**), where "S" is the subject and "P" are partners. **C** Rolling average of cooperation rate (green) and group size (red) as a function of trial for two sample participants. **D** Group level rolling average of cooperation rate (upper panel) and group size (lower panel) as a function of trial. Background lines represent 3 sample subjects with high, medium and low cooperation rates.

participants received the task instructions and watched a randomly chosen participant (in the current session) play approximately the first ten trials (up to the first stay/switch decision) as an experimenter-guided demo of the task. Participants were told that their social partners could break network links with them, but participants specifically were *not* instructed that maintaining network links was important to score or task performance. The timing of the start of the game play was synchronized so that each session groups' participants could feel that they were playing the game together. The session for the task began with the selection of a face of one's own avatar. There were four faces at the upper left corner of the screen. The participant pressed one of the four buttons corresponding to the four faces. After the selection of the face and the button press, one face remained on the screen as the self-avatar and the other three faces disappeared from the screen. Then, the main set of trials began (Fig. 1). During the pause before the first trial, there were no partners in the social network. The first newcomer appeared in the very first trial of the task and the participants played the PD task with this partner. Participants were not told how many exactly trials they would play.

**Prisoner's dilemma choice details.** The participants won 30 points when both they cooperated, and the partner cooperated. The payoff was 60 when the participants defected, and the partner cooperated. On the other hand, when the participants cooperated and the opponent defected, the participants lost 30 points. When both parties defected, the participants earned nothing. The points were accumulated across trials and the cumulative score was shown on the task screen (Fig. 1). The participants were told that the participation rewards were adjusted based on this total value. But the actual participation rewards were the same across the participants.

**Network link dynamics.** At the start of some trials (~10% of the trials), participants encountered a newcomer who automatically joins the network and becomes the current partner for that trial (Fig. 1). The participants then play the PD game with the newcomer in these trials, otherwise for trials without newcomers the current partner is randomly chosen from the existing network. At the end of some trials (~20% of the trials) after performing the PD game, participants decided whether to continue to play the game with the current partner in that trial or not (the stay/switch decision). The partner was given the same stay/switch decision and was more likely to switch if the participant was more defective recently. The social link was maintained if both chose to stay, while the link was broken if either partner or participant decided to break it. If the link broke, to "reset" future interactions with their session group, that partner's avatar was never used again (instead a new avatar face is assigned to subsequent newcomers). In addition to sequential newcomer events being forbidden, the probability of a newcomer event is reduced by half if the participant defects sequentially on the two immediately preceding trials (while the probability of partner switching stays the same). This rule is to prevent extreme defective behavior by making highly defective participants slowly trend towards having no partners over tens of trials, as reaching zero partners results in a short time-out period of a blank screen for several seconds before another newcomer is assigned. In terms of the bigger behavioral picture, when links are maintained, the interactions with specific social partners have to be memorized according to avatar faces in order to make adaptive decisions in the next encounter with the same social partner. No historical task summary for any partner is provided to the participants during the session.

**Session end conditions.** At the end of the session, we debriefed the participants about their general experience and to ensure they thought

they were playing with the other human participants in the experimental room. The feeling of deep engagement of participants during the task was visibly noticeable. Some participants squeezed their chairs, scratched their head aggressively, or cried out with frustration at times in response to events during their session. Despite this, reaction times were stable across the length of the experiment, not exhibiting any signs of speeding up or slowing down (Fig. S7). During the post-experiment interview, no participants had explicitly reported they were aware of playing the game against computer opponents.

**Algorithm of social partner.** One reason why the social relationship in the task was dynamic is that we programmed the "attitude" of the computer agent to change dynamically. This dynamic nature came from two sources in the agents' algorithm: one from the learning process provided by a reinforcement learning-like choice algorithm and another from cyclical alternations between the cooperative and non-cooperative strategies. The first mechanism was fairly predictable for the participants. If the participant chose cooperation more frequently, then usually the partners would be more cooperative for a time. But if the participant defected often, the partner would generally eventually tend to reciprocate either through defection or link breaking. On the other hand, partner dynamics were often unpredictable, due to a sinusoidal "mood" variable that was sampled semi-randomly (more technically, the mood variable was sampled by the absolute trial number from a sine wave with a period of ~22 trials for each partner, each partner had a different wave phase, and newcomer partners were randomly chosen out of a pool of 30 possible pre-initialized partners throughout the session). Thus, even if the participant kept a cooperative attitude toward the computer agents, a computer partner could suddenly exhibit a non-cooperative attitude (or vice versa). See *Data Availability* for the task script containing the partner algorithm.

Partners had a weak starting preference for defection to ensure cooperation was not artificially promoted (partners empirically *defected* on 58.2% of all newcomer trials, while human participants *cooperated* on 64.6% of all newcomer trials).

**Behavioral modeling**
For behavioral analysis we used linear mixed model approach (LMM), as implemented in the *lme4* R package. We built a total of eight models, results of each being represented in the panels of Fig. 2. Model structure was as follows:

1. *Effects of GS on cooperation:* choice type (cooperation/defection) ~ GS + (GS|participant)
2. *Effects of GS on reciprocity:* reciprocity (reciprocity/non-reciprocity) ~ GS + (GS|participant)
3. *Effects of ID on cooperation:* choice type (cooperation/defection) ~ ID + (ID|participant)
4. *Effects of ID on reciprocity:* reciprocity (reciprocity/non-reciprocity) ~ ID + (ID|participant)
5. *Effects of GS and cooperation on RT:* RT ~ GS * choice type(cooperation/ defection) + (1|participant)
6. *Effects of GS and reciprocity on RT:* RT ~ GS * reciprocity (reciprocity/ non-reciprocity) + (1|participant)
7. *Effects of ID and cooperation on RT:* RT ~ ID * choice type(cooperation/ defection) + (1|participant)
8. *Effects of ID and reciprocity on RT:* RT ~ ID * reciprocity (reciprocity/ non-reciprocity) + (1|participant)

Terms in brackets signify random effects; '~' separates the dependent variable from the predictors. Participants were treated as a random effect. The random effects structure was maximal (all possible random slopes and interactions), unless the model did not converge, in which case it was systematically simplified until convergence was achieved[31]. In the case of RT models, since some of them only converged with the simplest random structure (participant-specific random intercepts), we adopted the simple structure in all models, to ensure fair comparisons. Effects of group and

reciprocity were estimated separately due to high collinearity, which would affect reliable parameter estimates if used in the same model.

**Cognitive modeling**
Each model consists of 2 main sub-models, reflecting the *valuation* and *choice* functions. The valuation function quantifies the influence of relevant factors on the latent value of cooperation. The choice function determines the mapping between value and choice.

**Valuation function.** In its most general form, the model predicts choice type (cooperation *vs* defection) by modeling the latent *value of cooperation* $V^C$ on each trial $t$ as a linear function of the intrinsic value of cooperation $V_0^C$, and trial-specific values of reciprocation $V_t^R$, group size $V_t^{GS}$ and predicted cooperation $V_t^P$:

$$V_t^C = V_0^C + V_t^R + V_t^{GS} + V_t^P$$

$V_t^R$ is a difference of the terms representing value of reciprocating cooperation $V^{RC}$ and defection $V^{RD}$, which are weighted by the subjective probability of current partners (*i*) previous response being cooperation $p(P_iC_{-ID})$ and defection $p(P_iD_{-ID})$ respectively:

$$V_t^R = V^{RC}p(P_iC_{t-ID}) - V^{RD}p(P_iD_{t-ID})$$

Index *t-ID* represents the trial on which the previous interaction with the *i*th partner took place.

The probabilities $p(P_iC_{t-ID})$ and $p(P_iD_{t-ID})$ are modeled using an exponential decay function controlled by decay parameter $k \in [0, 1]$, representing memory decay rate as a function of interaction distance:

$$MR_t = Ik^{ID}$$

where $MR_t$ is the memory retention value at trial $t$, ranging from 0 (no memory) to 1 (perfect memory) and $I$ is the identity of partner's previous choice (fixed to 1). $MR_t$ is then transformed to probability space:

$$p(I) = 0.5(MR - 1) + 1;$$

$$\begin{cases} if\, P_iC_{t-back} = 1 & p(P_iC_{t-back}) = p(I);\ p(P_iD_{t-back}) = 1 - p(I) \\ if\, P_iC_{t-back} = 0 & p(P_iD_{t-back}) = p(I);\ p(P_iC_{t-back}) = 1 - p(I) \\ else & p(P_iC_{t-back}) = p(P_iD_{t-back}) = 0.5 \end{cases}$$

The transformation does not allow for the probability of true choice to decay below 0.5.

$V_t^{GS}$ is the product of a linear weight $V^{GS}$ and the group size at trial $t$:

$$V_t^{GS} = V^{GS} * GS_t$$

$V_t^P$ is the product of a linear weight $V^P$ and current partner's predicted cooperation probability:

$$V_t^P = V^P * pred_t^C$$

$pred_t^C$ is modeled using a simple reinforcement-learning (RL) delta-rule model, where participants estimate current partner's $P_i$ probability of cooperation $C_t$ as follows:

$$p(P_iC_1) = 0.5$$

$$p(P_iC_{t+1}) = p(P_iC_t) + \alpha PE;$$

$$PE = P_iC_t - p(P_iC_t)$$

Here, the prediction of cooperation of the *i*-th partner is initialized at 0.5 and updated after each choice *via* a prediction error *PE* (difference

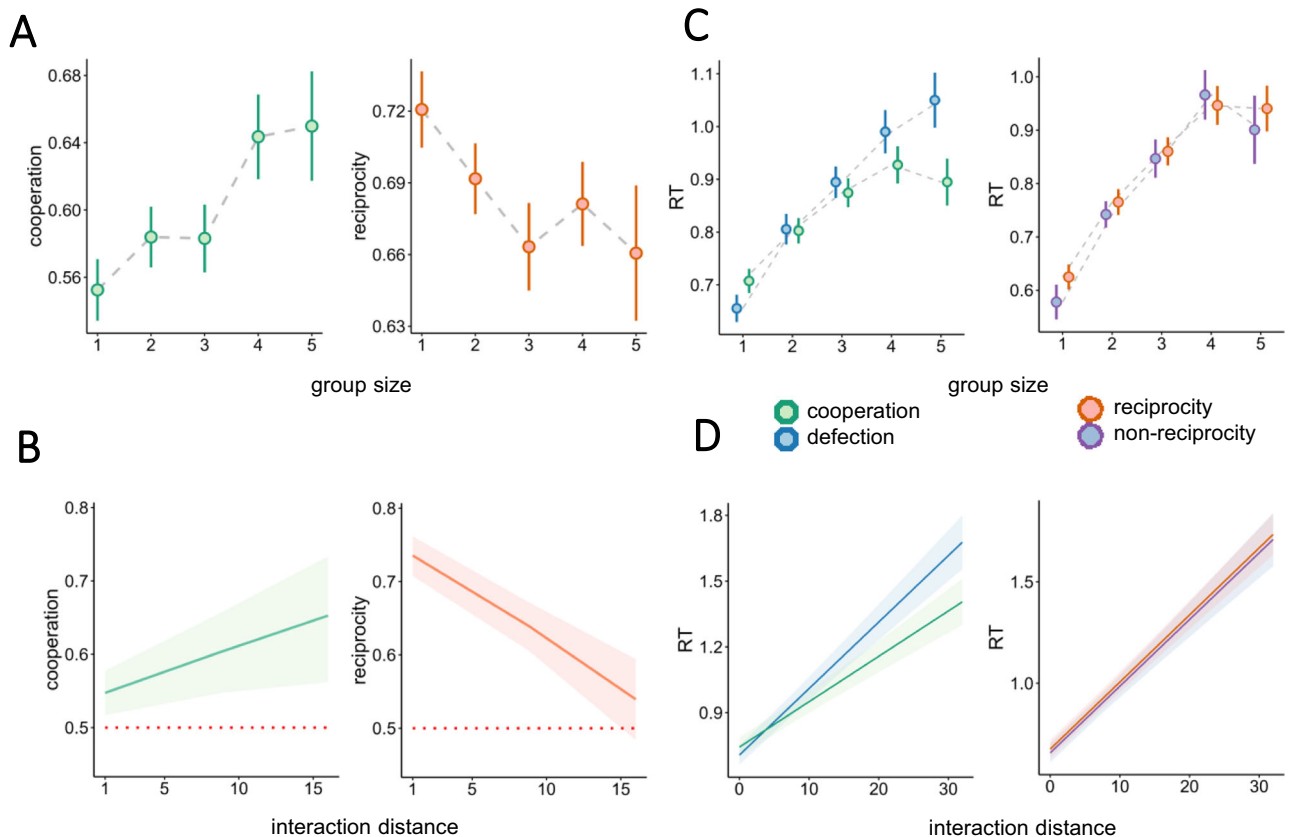

**Fig. 2 | Behavioral results.** Cooperation (left) and reciprocity (right) as a function of group size (**A**) and interaction distance (**B**). Red dotted line indicates chance level (50%). **C** Response times for cooperation (green) and defection as a function of group size (**C**) and ID (**D**). Error bars and ribbons represent 95% CI.

between partner's previous action $P_iC_t$, where cooperation = 1 and defect = 0 and the predicted probability of cooperation $p(P_iC_t)$) weighted by a learning rate $\alpha$. The component represents the value of predicted mutual cooperation, in line with research showing cooperation frequency is affected by the belief about partner's choice[32–34]. It rests on the assumption that people make inferences about their partner's likely action[35], and this prediction affects their strategy. In contrast to the probabilities of previous choice $p(PC_{t-1})$ $p(PD_{t-1})$, this value is not influenced by memory, but rather simple cashed values[36].

**Choice function.** The choice is modeled using the *softmax* function, where $V_t^C$ is compared to the value of defection, which is fixed to 0:

$$p(choice = Coop) = \frac{e^{-\tau V_t^C}}{e^{-\tau V_t^C} + e^{-\tau \cdot 0}}$$

Since the value of defection is fixed, this formulation is equivalent to using the logistic function, rendering the inverse temperature $\tau$ unidentifiable. Hence, $\tau$ was fixed to 1.

**Model variants.** We test two hypotheses, referred to as *increased prosociality* (P+) and *forgetting* (F+). P+ is formalized as $V^{GS} > 0$ (i.e., larger groups make people more prosocial by increasing the value of cooperation). F+ is formalized as $0 < k < 1$ (i.e., memory decay influencing the value of reciprocation, which has an *unmasking* effect on $V_0^C$). Based on these hypotheses, we test 6 models using a factorial design where we manipulate:
- group effect ($V^{GS} = 0$ *vs* $V^{GS}$ as a free parameter)
- decay effect ($k = 1$ (perfect memory) *vs* $k$ fixed =0 (no memory) *vs* $k$ as a free parameter).

**Model fitting.** We employ hierarchical fitting Hamiltonian Monte Carlo method, as implemented in the Stan Programming Language[37]. For each of the models, we generated 4 independent chains of 2000 samples (with 1000 burn-in) from the joint posterior distribution. Following recent recommendations[38], we use $\hat{R} < 1.04$ as a diagnostic of proper chain mixing, reflecting a reliable parameter estimate. $\hat{R}$ statistic reflects the proportion of between-to-within chain variance. We compare the models using the Leave-One-Out information criterion (LOOIC) and Widely-Applicable Information Criterium (WAIC) scores[39].

**Priors and parametrization.** We use a novel approach referred to as *beta-restricted priors*, where posterior parameter distributions have a constrained value range (i.e., explicitly specified minimal and maximal values) and a continuous distribution. This approach allows us to mitigate common issues with hierarchical Bayesian estimation, such as Neal's Funnel[40] (via the use of non-centered fitting) and poor identifiability[41] (via reducing the parameter ranges to plausible values). For this, we first define priors in '*raw*' space:

$$\alpha_p \sim \Gamma(1, 1)$$

$$\beta_p \sim \Gamma(1, 1)$$

$$p^{raw} \sim B(\alpha_p, \beta_p)$$

where $\Gamma$ is the gamma distribution, and $p^{raw}$ is an individual parameter of interest distributed in accordance with a beta distribution B, with shape parameters $\alpha_p$ and $\beta_p$. Then, these *raw* values are transformed to their native

space:

$$p = p^{raw} * \left( UB_p - LB_p \right) + LB_p$$

where $LB_p$ and $UB_p$ are the lower and upper parameter boundary, respectively. Similarly, we can compute the distributional statistics for the group-level parameter distribution by utilizing $\alpha_p$ and $\beta_p$. The posterior group mean is given by:

$$\mu_p = \frac{\alpha_p}{\alpha_p + \beta_p} * \left( UB_p - LB_p \right) + LB_p$$

Intuitively, this approach allows for fine control over parameter ranges, making it easier to avoid issues with parameter identifiability without the need to resort to highly improbable and non-informative uniform priors.

The following parameter ranges were used:

$$V_0^C \in (-10, \ 10)$$

$$V^{RC} \in (-10, \ 10)$$

$$V^{RD} \in (-10, \ 10)$$

$$V^{GS} \in (-5, \ 5)$$

$$V^P \in (-5, \ 5)$$

$$k \in (0, \ 1)$$

$$\alpha \in (0, \ 1)$$

The ranges of $V_0^C$ and $V^{RD}$ were set so that extreme values of cooperation and reciprocity were able to be fitted and recovered (Fig. S6). Ranges of $V^{GS}$ and $V^P$ were set so that they would allow for very strong effects of the linear predictors (up to 5 standard deviations above the mean, as all predictors were z-scored). The ranges of $k$ and $\alpha$ expressed natural parameter bounds. For identifiability purposes, $V^{RC}$ and $\tau$ parameters were fixed to 1 during model fitting.

## Model comparison

We compare model fits based on qualitative and quantitative measures. Qualitatively, a candidate model is required to reproduce main behavioral effects (Fig. 2A, B). Quantitively, we compare Leave-One-Out Information Criterion (LOOIC). LOOIC is a metric designed for hierarchical Bayesian models, which evaluates the model fit while considering model complexity, with lower values of LOOIC indicating better out-of-sample model prediction performance[39]. We also provide Widely-Applicable Information Criterion (WAIC) scores, which are an alternative method of quantitative model assessment, with slight differences with respect to the computation[39].

**Model recovery.** Following work using similar modeling framework[42,43] we tested the parameter recovery of the best fitting model by simulating independent 10 datasets (each consisting of the same number of trials and participants as the original design) and recovering parameter values by fitting the model (*Recovery of model parameters* and Fig. S6 in *Supplementary Materials*). For group-level parameters, we estimate whether the 90% Highest Density Interval (HDI) of the recovered posterior distribution contains the true parameter value (should hold true for 90% of the intervals, on average). For participant-level parameters, we correlate

the recovered posterior means with simulated values, per simulation (Fig. S5).

**fMRI data acquisition and preprocessing.** fMRI scanning was performed on a 3 T Siemens Magnetom Prisma scanner and a 43-channel head coil at the CiNET facility in Osaka, Japan. Functional whole-brain images we acquired using an echo-planar imaging (EPI) sequence with the following parameters: repetition time (TR) = 2000 ms, echo time (TE) = 30 ms, flip angle = 75°, field of view (FOV) = 200 mm, slice thickness = 2 mm, voxel size = $2 \times 2 \times 2$ mm$^3$, multiband acceleration factor of 3, gap = 0 mm, ascending interleaved slice acquisition of 72 axial slices. High-resolution T1-weighted anatomical scans were acquired using an MPRAGE pulse sequence (TR = 1900 ms, TE = 3.37 ms, FOV = 256 mm, image matrix $256 \times 256$, slice thickness = 1 mm).

Preprocessing and analyses were performed using SPM12. For each participant, the raw images were slice-time corrected, aligned to the first volume to correct participants' head motion, spatially normalized into the Montreal Neurological Institute (MNI) template and smoothed using a 4-mm full width at half maximum Gaussian kernel. All images were temporally filtered using a high-pass filter with a width of 128 s.

**Statistical fMRI analysis.** We constructed five GLM models to map model-based parameter values from the best fitting cognitive model to brain activity. All models had similar structure, such that the BOLD response in each voxel was predicted by the events (coded as stick regressors) related to: new partner introduction (10% of trials), choice, choice feedback, reward screen, switch (20% of the trials) and switch feedback (20% of trials). Additionally, all models contained parametric regressors of RT and memory decay parameter $MR_t$ during choice period, a pause regressor (which reflected few-trial long pauses affecting participants losing all their partners, until a new one is introduced), and 6 movement-related regressors. All models introduced additional parameters reflecting model-based estimates, coded as parametric regressors of the choice period. GLM 1 tested the integrated value of cooperation $V_t^C$. GLM 2 tested a transformation of $V_t^C$ measuring integrated value of reciprocity $\bar{R}_t$:

$$\bar{R}_t = \begin{cases} PC_{t-ID} = cooperation & V_t^C \\ PC_{t-ID} = defection & -V_t^C \end{cases}$$

GLM 3 tested a $V_t^C$ transformation with respect to the integrated value of forgiveness $\overline{NR}_t^F$ (non-reciprocal trials where participants cooperated after their current partner's defection):

$$\overline{NR}_t^F = \begin{cases} PC_{t-ID} = cooperation & 0 \\ PC_{t-ID} = defection & V_t^C \end{cases}$$

and with respect to the integrated value of betrayal $\overline{NR}_t^B$ (non-reciprocal trial where participant defected after partner's cooperation):

$$\overline{NR}_t^B = \begin{cases} PC_{t-ID} = cooperation & -V_t^C \\ PC_{t-ID} = defection & 0 \end{cases}$$

GLM 4 was constructed for a PPI analysis (see: *PPI* section) with a seed within the precuneus. It contained all regressors of GLM 1, together with the BOLD timeseries extracted from the seed region and all seed-regressor interaction terms.

GLM 5 was constructed to test regions sensitive to congruence between social tendency, and ID-moderated reciprocal option. Social tendency $ST_i$ defined as a binary variable, reflecting whether participant $i$ was, on average, more likely to cooperate or defect:

$$ST_i = \begin{cases} 1 \ if \ p(C_i > 0.5) \\ 0 \end{cases}$$

where $p(C_i)$ is the overall probability of cooperation.

*Congruence* $CG_t$ was defined as a conjunction between $ST_i$ and current partner's previous choice $PC_{t-ID}$, such that:

$$CG_t = \begin{cases} 1 \; if \; ST_i \; = \; PC_{t-ID} \\ 0 \end{cases}$$

where $PC_{t-ID} = 1$ indicates cooperation, and $PC_{t-ID} = 0$, defection.

GLM 5 contained the same regressors as GLMs 1-3, with parametric modulation of RT, *congruence*, and binarized ID, where *short* distance was defined as exactly ID = 1, and *long* distance was defined as ID > 1.

**ROI analysis.** All ROIs were independently defined based on an online meta-analysis using keywords in the *Neurosynch*[44] database. The keywords were dictated by the hypothesized function of these areas within the context of our task. In order to obtain a reasonable sample size of studies, the keywords were as general as possible. Additionally, the activity clusters needed to be located within the anatomical bounds of the region for which the ROI was being obtained. All ROIs were spherical, with varying volumes. The volumes of the ROIs were dictated by a) anatomical size of the structure of interest, and b) the size of the functional cluster, within the arbitrary limits of not being smaller than 5 mm$^3$, and not larger than 10 mm$^3$. ROI analysis was performed by extracting whitened and filtered voxels within an ROI from a contrast of interest and averaging the voxel values within participants. The vector of averaged beta weights was then compared against 0 using a t-test.

**Value of cooperation.** The VMPFC ROI was defined as 10 mm$^3$ sphere, centered at the peak activity for the keyword '*value*' (peak MNI coordinates: x = 0, y = 40, z = -8; 407 studies included in the database). The choice of VMPFC was driven by the *a priori* expectation of the region being involved in subjective value processing[45] (including social contexts[46]).

**Forgetting.** This set of ROIs was chosen in order to test the connectivity between memory representation (precuneus) and value representations, and included: precuneus, FFG, NAcc and dACC.

The precuneus ROI was defined as 10 mm$^3$ sphere, centered at the peak activity for the keyword '*memory*' (peak MNI coordinates: x = -8, y = -66, z = 28; 2744 studies included in the database). The choice of precuneus was motivated by both: an *a priori* expectation of the region playing a crucial role in social memory[47] as well as the results of earlier model-based analyses, relating its activity to forgetting.

The FFG ROI was defined as two 6 mm$^3$ spheres, centered at the peak activity for the keyword '*face recognition*' (peak MNI left: x = -42, y = -52, z = −20; peak MNI right: $x = 41$, $y = −52$, $z = −20$; 79 studies included in the database). The choice of FFG was motivated by both: an *a priori* expectation of the region being involved in face recognition and memory[48,49], as well as the results of earlier model-based analyses, relating its activity to forgetting.

The NAcc ROI was defined as two 5 mm$^3$ spheres, centered at the peak activity for the keyword '*value*' (peak MNI left: $x = -6$, $y = 10$, $z = −5$; peak MNI right: $x = 6$, $y = −10$, $z = −5$; 407 studies included in the database). The choice of NAcc was motivated by both: an *a priori* expectation of the region being involved in reward processing[50], as well as the results of earlier model-based analyses, relating its activity to the value of cooperation.

The dACC ROI was defined as a 10 mm$^3$ sphere centered at the peak activity for the keyword '*control*' (peak MNI: $x = 4$, $y = 18$, $z = 42$; 3796 studies included in the database). The choice dACC was motivated by both: an *a priori* expectation of the region being involved in integrating value information across timescales[51,52], as well as the results of earlier model-based analyses, relating its activity to the value of reciprocity.

**Congruence.** For the congruence analysis, we re-used the precuneus and dACC ROIs defined above. Additionally, we built 2 ROIs for the left and right DLPFC, defined as 5 mm$^3$ spheres, centered at [−40, 40 24] and [40,

40 24], respectively. Similarly, to the earlier definition of dACC, we used *Neurosynth* peak coordinates related to the term '*control*', within clusters which anatomically included the DLPFC region. We chose those regions for both theoretical and practical reasons (i.e., significant whole-brain activity related to key functional roles in prior GLM models). The DLPFC and dACC have been associated with processing value-based information across different timescales[51,52], while the precuneus is a central hub of the mentalizing network, suggested to play a role in social working memory[47,53].

*PPI*

We performed an ROI-based PPI analysis to test the connectivity between *a)* precuneus and fusiform gyri, and *b)* precuneus and NAcc, and *c)* precuneus and dACC during the choice period, as a function of memory retention $MR_t$. The psychophysiological interaction (PPI) analysis was performed using the gPPI (generalized PPI) toolbox[54], which explicitly models the whole task on top of the PPI-specific regressors. We tested our hypotheses by testing the connectivity of the seed with *a)* fusiform, *b)* NAcc, and *c)* dACC, as a function of the $MR_t$ regressor.

## Reporting summary

Further information on research design is available in the Nature Portfolio Reporting Summary linked to this article.

## Results

On each trial, participants interacted with one member of their group chosen randomly. The size and composition of the group changes dynamically, as every few trials either a new partner was introduced (start of ~10% of trials), or both agents were given a choice to exclude the current partner from their group (end of ~20% of trials; Fig. 1A, B; task video in *Supplementary Materials*). All group sizes were experienced fairly consistently across the experiment (Fig. 1D). Trial payoff was determined by a PD payoff matrix (Fig. 1A). Participants were rewarded based on the total accumulated score.

## Cooperation and reciprocity are influenced by group size and interaction distance

Overall, cooperative behavior was dominant (57% of the trials, across-participant SD = 14%). Importantly, cooperation was modulated by group size (GS) β = 0.11 95% CI [0.07, 0.15], $p < 0.001$ and interaction distance $p < 0.01$ (ID; defined as the number of trials since last interaction with the current partner) β = −0.06 95% CI [−0.01, −0.10] $p < 0.01$ (Fig. 2A). Reciprocity was defined as replicating the previous choice of the current partner. Across the session, reciprocal behavior was also dominant (70% of the trials, across-participant SD 12%). Importantly, it was negatively modulated by GS β = −0.06 95% CI [−0.11, −0.02] $p < 0.01$ and ID β = −0.10 95% CI [−0.15, −0.05] $p < 0.001$. (Fig. 2B).

Response times (RTs) were longer for cooperative (M = 796 ms) as compared to defective choices (M = 760 ms) β = 89.5, 95% CI [62.7, 116.2] $p < 0.001$, and increased as a function of GS β = 87.1 95% CI [78.6, 95.4] $p < 0.001$ as well as ID β = 30.4, 95% CI [26.5, 34.2] $p < 0.001$. The interaction between GS and choice type revealed that the RT increase as a function of GS was faster for defection β = −36.4 95% CI [−46.5, −26.3] $p < 0.001$. Similarly, interaction between ID and choice type indicated a larger RT increase for defection β = −9.6 95% CI [−14.6, −4.68] $p < 0.001$. No significant differences were found between reciprocal and non-reciprocal choice RTs (Fig. 2C, D).

These results suggest that the trade-off between cooperating and reciprocating of partners' actions was systematically modulated by groups size and interaction distance (see Figs S3, S4 and Tables S1–S3 for complimentary analyses). To understand the nature of this relationship, we performed a model-based analysis.

## Estimating the value of cooperation

We aimed to formally quantify the latent mechanisms driving choices on a trial-to-trial basis using computational modeling, implemented using

hierarchical Bayesian methods. We assumed participants make noisy value-based choices, where values of cooperation and defection are determined by a set of variables, which include the value of reciprocation, baseline cooperation level (*social tendency*), as well as trial-by-trial predictors including partner-specific history of interactions (see: *Methods*). We tested two alternative mechanisms which can explain the observed pattern of results: group size-induced increased prosociality (*P+*) and distance-related forgetting (*F+*). *P+* assumes that the value of cooperation is directly influenced by group size, so that people become more prosocial as the group gets larger, e.g. from increased peer pressure or positive feelings related to being in a larger group[55]. *F+* assumes an indirect effect of forgetting cooperation, quantifying the intuition that using a reciprocal strategy becomes less appealing if participants cannot be certain which previous action the partner had taken to reciprocate. Specifically, we assume that the memory fidelity of the current partner's previous choice decays exponentially as a function of ID. By reducing the value of reciprocation, the relative social tendency now dominates the choice function, making the prediction that worse memory should lead to an increase in the individual default tendency (either cooperation or defection, depending on the participant).

## Value of cooperation dependents on forgetting
We tested six model variants by crossing factors representing increased prosociality (present or absent) and memory retention (perfect, decaying, or none; Fig. 3A, B). As expected, only models containing increased prosociality or forgetting parameters could predict the qualitative patterns of increased cooperation and decreased reciprocity as a function of group size observed in the data. Quantitative model comparison using leave-one-out cross-validation information criterion (LOOIC) scores[39] revealed that the model with decaying memory but without increased group size-induced prosociality explains the data in the most parsimonious fashion (LOOIC of 13,777 versus 13,788 for the second-best model; Fig. 3C, Table 1), providing evidence for *F+*, but against *P+* hypothesis. Posterior predictive checks indicated that the winning model provided a good fit to the data for most of the participants (Fig. 3E–G). Group mean of the parameter $k$ representing memory decay was estimated at 0.638 95% CI [0.579, 0.696], suggesting that, on average, memory trace of partner's choice was almost entirely gone after around 10 trials back (Table 2; Fig. 3D).

## Brain signatures of model-derived measures
To explore brain correlates of choices, we regressed predictions derived from the winning model to the brain activity, as measured by the blood-oxygen-level-dependent (BOLD) fMRI signal. A standard general linear model (GLM 1) analysis revealed that value of cooperation $V_t^C$ was associated with the activity within the ventral striatum/nucleus accumbens area (VS/NAcc; peak MNI left: $[-8, 2, -10]$; right: $[8, 0, -6]$). Since $V_t^C$ represents subjective value, we also expected to see signals related to it within the ventromedial prefrontal cortex (VMPFC), area heavily implicated in processing subjective value signals[45]. To test this, we performed an analysis using an ROI defined from Neurosynth[44] meta-analysis using the term '*value*' (*Methods*; for a similar approach, see ref. 56). In line with this prediction, we found $V_t^C$ to significantly predict activity within the VMPFC ROI t(24) = 2.592 CI [0.0368, 0.3243] $p$ = 0.016 (Fig. 4A). While there were no areas predictive of reciprocity, GLM 2 model revealed distinctive patterns predicting non-reciprocal ($\overline{NR}_t = -\overline{R}_t$) choices within the dorsal anterior cingulate cortex (dACC; peak MNI: [6 22 44]) and bilateral anterior insula (AI; peak MNI left: $[-32 \ 18 \ -4]$; right: $[24 \ 14 \ -16]$; Fig. 4B). Model-based prediction of forgiveness (GLM 3) was associated with a widespread brain activity including dorsomedial prefrontal cortex, broad dorsolateral activity, extending from the middle frontal gyri anteriorly [-22 -2 44], up to superior parietal lobule posteriorly (left: [-22 -56 34]; right: [28 -50 42]), as well as clusters within the lateral cerebellum $[-49 \ -56 \ -32]$ (Fig. 4C; Table S4). In contrast, effect of betrayal, $\overline{NR}_t^B$, was localized specifically to the bilateral dorsolateral cortex (DLPFC; left: $[-36 \ 22 \ 14]$; right: [40 48 24]).

## Neural correlates of forgetting
Next, we looked for areas associated with forgetting, quantified as:

$$F_t = 1 - k^{ID_t}$$

where $k$ is the participant-specific memory decay parameter, ranging from 0 to 1. This contrast revealed activity within the precuneus $[-12 \ -66 \ 30]$ and bilateral fusiform gyri (FFG; left: $[40 \ -52 \ -22]$; right: [-38 -48 -22]). We found no areas involved in the opposite contrast related to stronger memory retention.

## Functional connectivity between key nodes predicts memory retention
These results raise the question of how memory information flow is integrated into the brain and how it affects the integrated value of cooperation $V_t^C$. We were particularly interested in the coupling between (1) precuneus and the FFG, as a potential mechanism for retrieving face information encoded within FFG, (2) precuneus and VS/NAcc (cooperation value signal), and (3) precuneus and dACC (reciprocity value signal), which could reflect how the memory information is integrated into subjective value. We hypothesized the precuneus to be the central node of this information transfer due to its strategic location as a densely connected functional hub[53], as well as its relation to episodic memory retrieval[57-59], especially in social contexts[47]. To test this, we build a generalized PPI model (GLM 4) with the seed being placed within the precuneus. Similarly to the VMPFC ROI analysis, the 3 new ROIs were created independently from meta-analysis keywords, using the *Neurosynch* database (see *Methods*). We found $F_t$ to be associated with decreased connectivity between the precuneus and FFG t(23) = $-2.071$ CI $[-2.331, -0.003]$ $p$ = 0.050, VS/NAcc t(23) = $-2.230$ CI $[-2.982, -0.112]$, $p$ = 0.035, and a non-significant trend for dACC t(23) = $-1.95$ CI$[-2.257, 0.064]$, $p$ = 0.06 (Fig. 5B).

## Trade-off between cooperation and reciprocity reflected within the left DLPFC and dACC
We next searched for regions sensitive to the trade-off between cooperation and reciprocity, which drives choices in our cognitive model. For this, we first defined social tendency, i.e., whether a given participant was, on average, more likely to cooperate, or defect across a session. This empirically-derived value corresponded closely with model-derived values of the $V_0^C$ parameter (Pearson's $r$ = 0.72, t(23) = 5.11, $p < 0.001$). If a participant's cooperative tendency was consistent with the current partner's previous choice, the trial was labeled as *congruent*. By this logic, incongruent trials were ones where the behavioral tendency and the reciprocal option were opposite. In line with the modeling, we further expected this conflict to be moderated by ID, which we binarized by whether the previous encounter occurred on previous trial (ID = 1; short distance) or not (ID > 1; long distance). Whole-brain analysis found a cluster sensitive to this interaction effect within the left DLPFC [-26 40 16] (Fig. 6A). This area was also directly associated with the strength of social tendency between participants $r$ = 0.482, t(23) = 2.65, $p$ = 0.015.

We additionally tested a set of independently defined ROIs from candidate regions which we a priori hypothesized to potentially play a role, including the dACC, DLPFC and the precuneus. The DLPFC and dACC have been implicated in representing value-based information across different timescales[51,52]. The precuneus is a central hub of the mentalizing network, suggested to play a role in social working memory[47]. Among those, significant effect of the interaction was observed within the dACC $F(1,24)$ = 4.56, $p$ = 0.043 and left DLPFC ROI $F(1,24)$ = 5.87, $p$ = 0.023. Simple effects analysis within left DLPFC and dACC indicated that congruence was different only at short distance (DLPFC: t(24) = $-2.403$, CI $[-1.101, -0.083]$ $p$ = 0.023; dACC: t(24) = $-2.076$, 95% CI$[-0.803, -0.002]$, $p$ = 0.048), suggesting that memory-based uncertainty attenuated the conflict (Fig. 6B). Similar to the functionally defined left DLPFC, independently defined dACC and left DLPFC ROIs were also related to

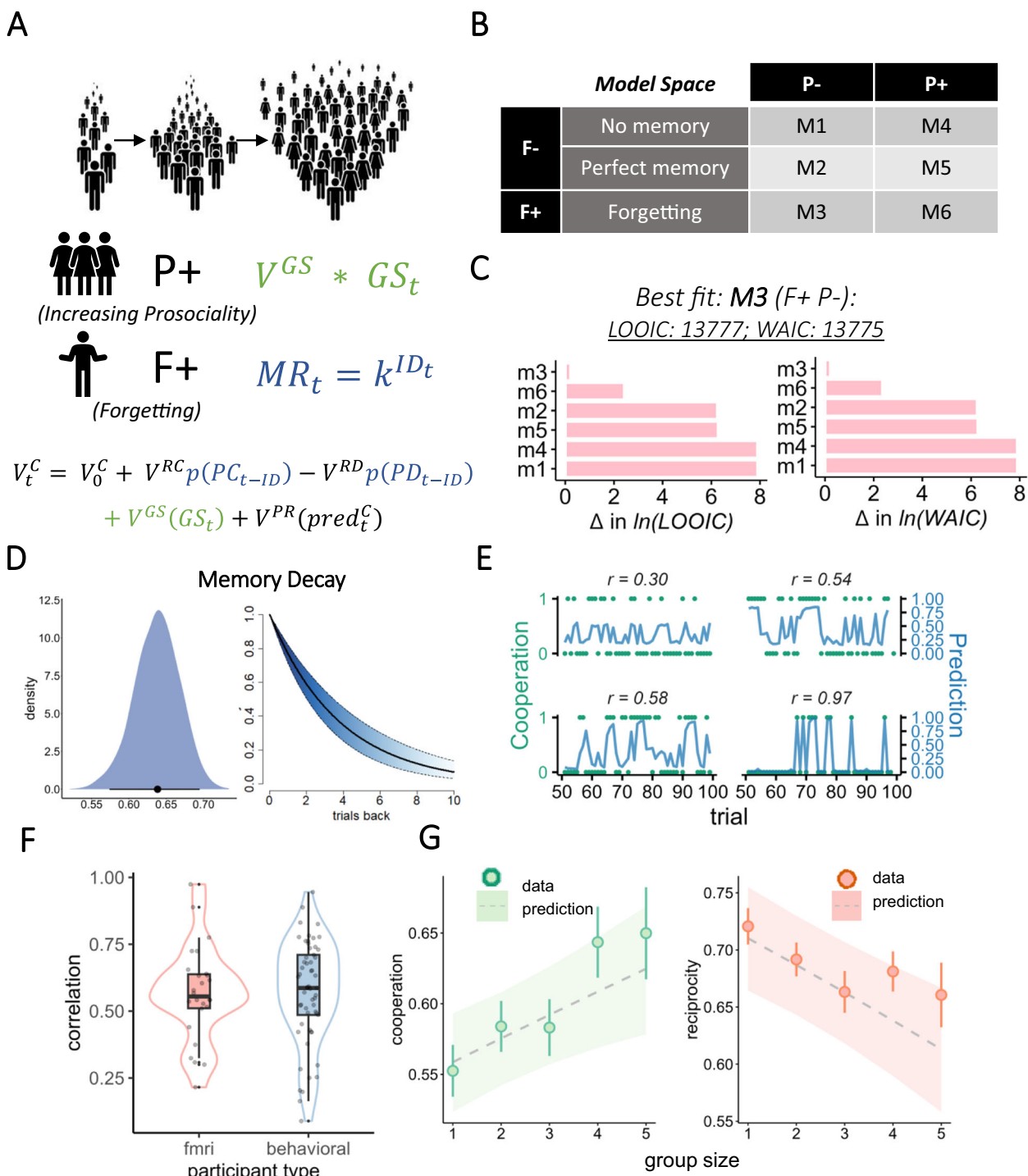

**Fig. 3 | Cognitive modeling. A** Hypothesis space, aimed at explaining why people cooperate more in larger groups. *Increasing prosociality* hypothesis (P+) assumes a linear increase in cooperation as a function of group size. *Forgetting* hypothesis (F+) assumes an exponential decay in memory retention (MR) of partner's previous choice as a function of interaction distance, which then influences the value of reciprocation. Bottom row represents the full value function, color-coded in line with the hypotheses. **B** Tested model space. **C** Model fit scores based on LOOIC (left) and WAIC scores (right). For clearer visualization, the scores were relativized with respect to the winning and plotted on a natural logarithmic scale. Vertical red line

represents 8-point difference, which signifies a significantly better fit. **D** Posterior distribution of the memory decay parameter *k* (left) and the predicted memory decay as a function of ID (right). **E** Winning model choice predictions (blue lines) vs observed choices)green dots) for 4 participants on a subset of trials. Pearson correlation coefficients r in the subplot titles relate to overall choice-prediction correlations for given participant. **F** Individual correlations between predicted and actual cooperative choices for 2 subsamples of participants. **G** Simulated model predictions (ribbon, width representing 95% CI) for cooperation and reciprocity, plotted against empirical values estimated from the data (error bars represent 95% CI).

**Table 1 | LOOIC and WAIC model scores. M1: no memory, no group effect; M2: perfect memory, no group effect; M3: imperfect memory (forgetting), no group effect; M4: no memory, group effect; M5: perfect memory, group effect; M6: imperfect memory (forgetting), group effect**

|     | LOOIC    | WAIC     |
|-----|----------|----------|
| M1  | 16498.99 | 16497.92 |
| M2  | 14295.95 | 14294.71 |
| M3  | 13777.47 | 13775.46 |
| M4  | 16481.08 | 16479.37 |
| M5  | 14310.33 | 14308.46 |
| M6  | 13787.56 | 13784.84 |

M1: no memory, no group effect; M2: perfect memory, no group effect; M3: imperfect memory (forgetting), no group effect; M4: no memory, group effect; M5: perfect memory, group effect; M6: imperfect memory (forgetting), group effect.

**Table 2 | Descriptive statistics of group-level parameter posterior distributions of the best fitting model; sd: standard deviation; q05: 5th quantile; q95: 95th quantile**

|           | Mean  | Median | sd     | q05   | q95   |
|-----------|-------|--------|--------|-------|-------|
| $V_0^C$   | 1.366 | 1.357  | 0.208  | 1.028 | 1.719 |
| $V^{RD}$  | 3.481 | 3.491  | 0.310  | 2.963 | 3.967 |
| $k$       | 0.638 | 0.638  | 0.0306 | 0.585 | 0.686 |
| $V^P$     | 2.536 | 2.540  | 0.220  | 2.166 | 2.901 |
| $\alpha$  | 0.567 | 0.567  | 0.042  | 0.499 | 0.636 |

social tendency across participants (DLPFC: $r = 0.390$, $t(23) = 2.03$, $p = 0.05$; dACC: $r = 0.390$, $t(23) = 2.03$, $p = 0.05$).

## Discussion

We show that dynamic changes in group size affect cooperative choices and brain activity via the indirect effects of partner interaction distance across time, in naturalistic groups where sequential pairwise interactions are dominant. Dyadic interactions are prevalent in our everyday lives, and their dynamics are crucial for the functioning of small groups, such as research groups, sports teams, corporate teams or music bands.

### Memory-based reciprocity

Specifically, our model predicts that the memory trace of the current partner's previous choice decays with time, affecting the reliability of reciprocate choices and biasing the response towards the dominant social tendency (cooperative or defective, participant-dependent). Such a mechanism requires recalling information regarding the partner's choice from the previous interaction (which could have occurred recently, or many trials ago), weighting it by subjective certainty, and integrating it with the value of one's social tendency (the last being a participant-specific "session-level" variable that could be influenced by individual differences as well as task design). Implementing this strategy requires sensitivity to information across varying timescales. The value of reciprocity needs to be dynamically evaluated at each trial, updated by information from a variable number of trials ago (dependent on ID), and weighted against the more stable, time-independent value of social tendency ($V_0^C$ in the best performing model). Converging evidence suggests that such cross-timescale computations are necessary for many, if not most, natural behaviors[60]. For example, frontal cortex neurons in marmosets predict social behavior multiple seconds prior to engagement, likely representing general social arousal[61], drawing a parallel to the biasing effect of social tendency in our study. In more controlled experimental conditions, both monkeys[62] and humans[63] track uncertainty across timescales to guide exploratory choices. This account is supported by

evidence of maintenance and utilization of uncertainty by working memory[64], reflected at the neural circuit level in the neuronal capacity of representing information across a plethora of time constants[52].

### Group size and cooperation

Group size plays a role in brain function and cooperative behavior across many timescales, ranging from seconds to years. Long-lasting group size changes are coupled with changes in brain structure and function[11], which reflect increased capacity for coordinating complex cooperative behaviors[65]. In contrast, short-term influences are highly context-sensitive. Depending on a range of crucial parameters, such as time horizon[66], dynamicity of the environment[24,25], the incentive structure[29], or whether within-group interactions are simultaneous or pairwise[67], group size can either decrease or increase cooperation. Our results reveal that how group size influences cooperative tendencies in iterative paradigms is indirect, influencing behavior through memory capacity limitations though still being based on the innate human desire for fairness and reciprocity. This finding suggests that processes leading to higher levels of cooperation are timescale-dependent, with the ability of keeping track of previous interactions being the linchpin, and preservation of fairness and reciprocity being a key motivation. Increased memory demand biases choices towards the default social tendency in the multi-trial, session-level time scales in our task, and likely acts as a selective force across lifetimes and evolutionary timescales as a core mechanism underlying group cooperation.

### Neural representations of value of cooperation and reciprocity

We find evidence for the integrated value of cooperation within NAcc/VS and VMPFC, areas associated with reward processing and subjective value[50,68], as well as mutual cooperation[69]. The value of non-reciprocity is associated with dACC and bilateral insula, areas implicated in the detection of unfairness and social injustice[70,71]. Interestingly, overriding reciprocal choice activity pattern manifests differently for the two types of non-reciprocity: in the case of forgiveness, we observe a widespread activation within medial (dACC) as well as dorsal fronto-parietal area, extending from the middle frontal, through precentral, back to bilateral supramarginal gyri, while activity related to betrayal (defection after partner's cooperation) is highly specific to the DLPFC. This discrepancy might reflect a tendency for non-reciprocity to require a stronger brain signature if it is overriding a baseline human tendency towards fair reciprocation, or alternatively taking longer to process previous partner defection[72].

### Neural correlates of forgetting

The forgetting parameter is related to activity of the bilateral FFG and the precuneus. FFG plays a direct role in face recognition[48] and memory[49]. Its sensitivity to distance might reflect modulation by face novelty (higher response to faces already slightly forgotten), or higher effort needed to accurately associate the face with previous interaction outcomes. The precuneus is a crucial node in the mentalizing network implicated in social working memory[47], and plays a major role in brain state transitions generally[57]. Its activity pattern also reflects the amount of effort exerted in social contexts[47], making it an appropriate medium for estimating memory certainty. Functional connectivity between FFG and precuneus, and precuneus and NAcc increased with greater memory retention, suggesting the influence of memory on the integrated value scales with memory certainty, as predicted by the model.

### Crucial roles of DLPFC and dACC in information integration across timescales

Finally, we identified left DLPFC and dACC as areas involved in integrating the long-term social tendency with the reciprocity context (current partner's previous choice, modulated by memory certainty). For both areas, activity was higher during incongruent trials (where the dominant social tendency is opposite to the reciprocal choice), but only at short interaction distance (where memory certainty was highest). Both areas were also sensitive to the absolute value of social tendency, coding the participant-specific preference

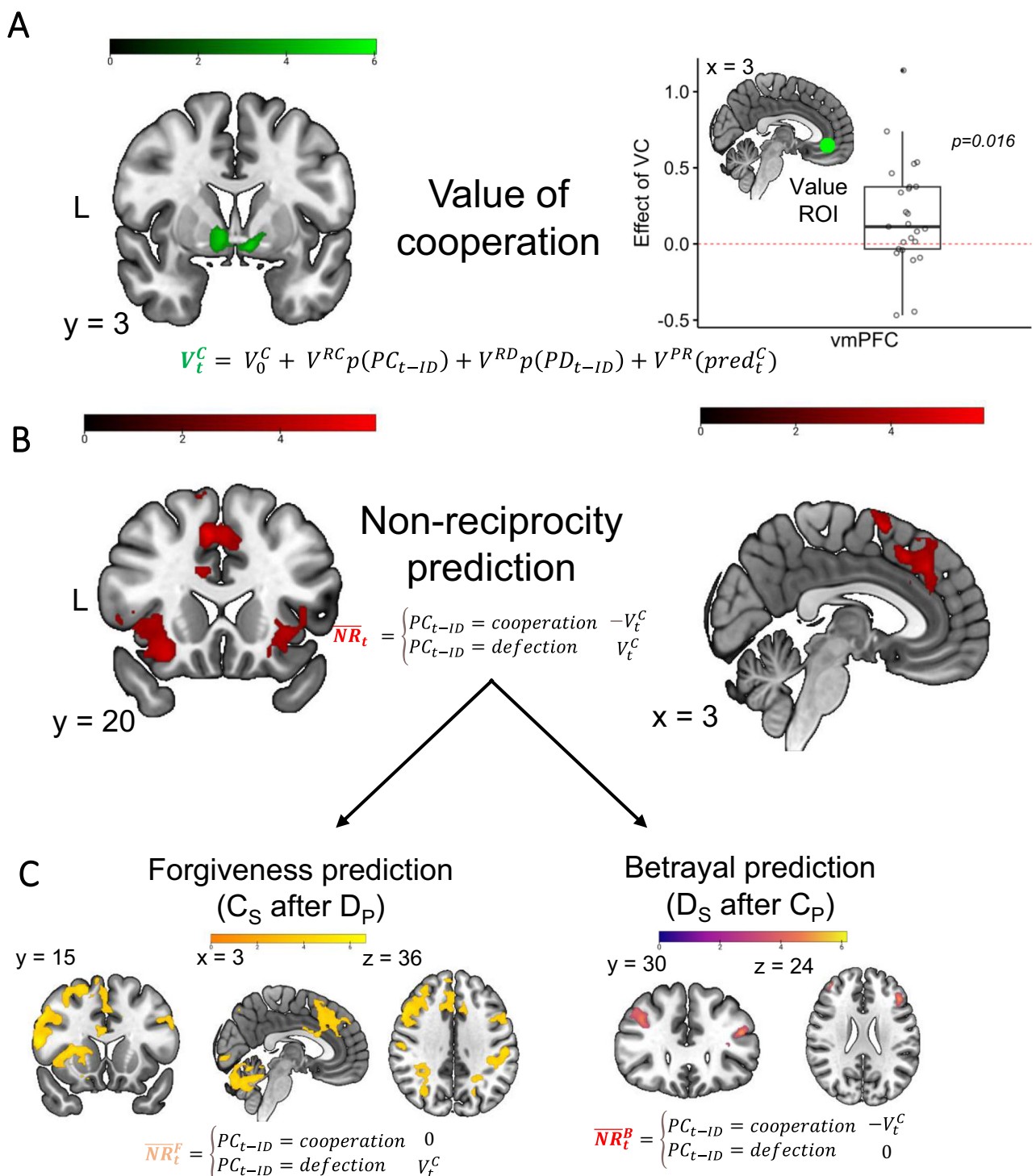

**Fig. 4 | Brain representations of model-based predictions. A** Value of cooperation was predictive of activations within the ventral striatum/Nucleus Accumbens (whole-brain analysis) and theoretically-derived ROI associated with value within the vmPFC. **B** Model-derived value of non-reciprocity was associated with activity within the bilateral AI and dACC. **C** Prediction of forgiveness (left) was associated with wide-spread activity across dorsolateral and medial areas, as well as the cerebellum. Prediction of betrayal was associated with bilateral activity within the DLPFC. All whole-brain analyses were conducted using a $p < 0.05$ cluster-wise family-wise error rate (FWE)-corrected, cluster forming threshold $p < 0.001$.

magnitude for cooperation or defection. dACC's involvement is consistent with its role in tracking and comparison of reward-relevant information across different temporal intervals[51]. As such, its activity has been proven critical for optimizing reward expectation in dynamic environments. Extending this functionality to the social domain, it allows for flexible value

adjustments by incorporating time-sensitive interaction history context into the value estimation. DLPFC can represent information on a wide range of timescales, from single to multiple trials back in the session history[52,73]. As such, it is well suited for functions related to the preservation of behavioral consistency, such as maintaining intentions[74], as well as dynamic control,

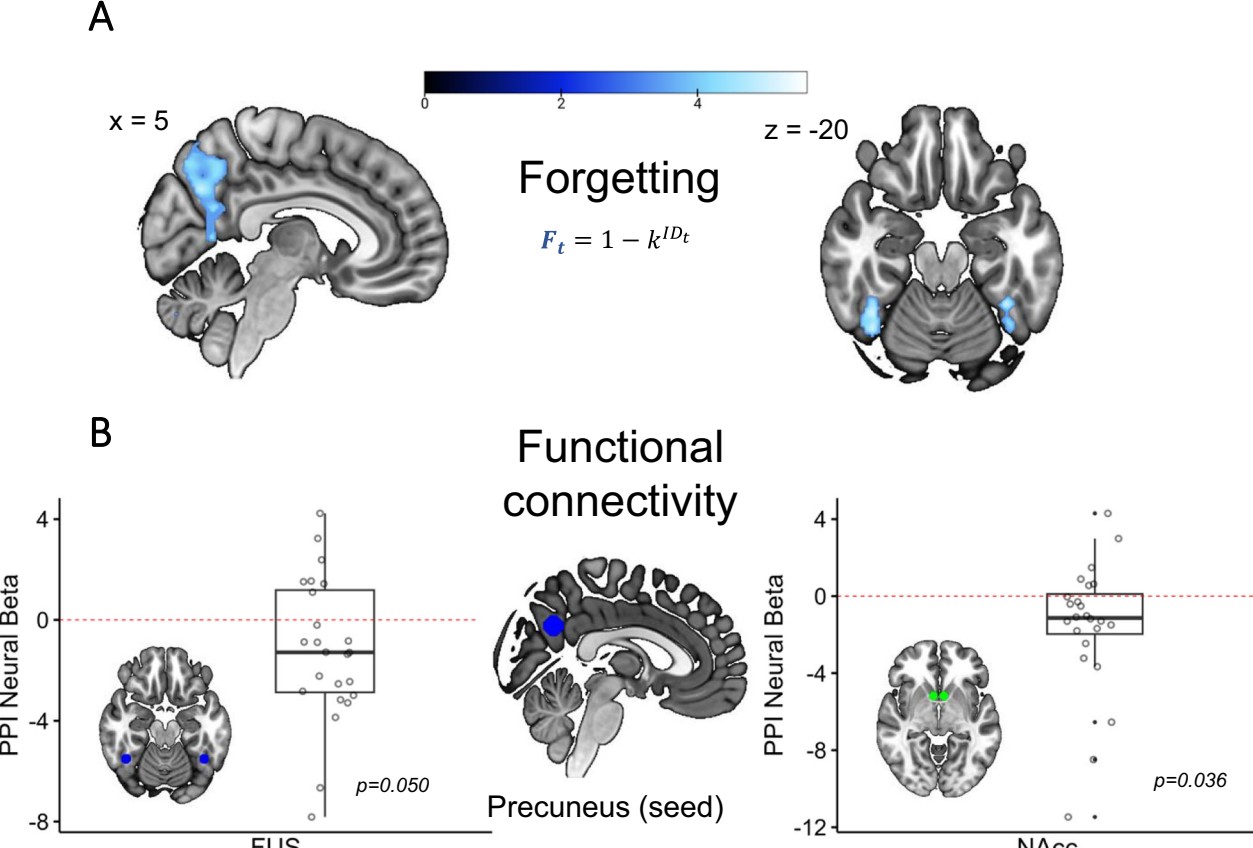

**Fig. 5 | Neural correlates of forgetting. A** Whole-brain analysis revealed precuneus and bilateral fusiform gyri to be associated with memory decay $p < 0.05$ cluster-wise FWE corrected, cluster forming threshold $p < 0.001$. **B** Connectivity between precuneus (seed) and two ROIs of interest: bilateral fusiform (left) and NAcc (right). Each point represents participant-specific connectivity strength decrease as a function of $F_t$, averaged across ROI voxels.

like reappraisal[75] or cognitive resource allocation[76]. In social game settings, DLPFC has been associated with exerting cognitive control in service of goal-directed behavior, necessary in situations of high cognitive load[75,77], as well as overriding default behavioral tendency[78]. Overall, the ability to integrate information from different temporal scales is crucial for computing value signals, where the value of reciprocation is moderated by the ID-dependent context.

Activity within the left DLPFC and dACC reflects not only the context itself (current partner's previous choice), but also its reliability, making the signal sensitive to all three crucial variables driving choice (social tendency, value of reciprocation, and memory retention). Interestingly, both of those areas were previously implicated with exerting cognitive control *via* the inhibition of tempting responses[79,80]. Our results suggest that their role might be more general, reflecting the temporal integration of value-relevant signals representing conflicting behavioral tendencies. Previous literature has also linked the DLPFC to functions that are conceptually related to betrayal, such as cognitive control[75] and modulation of social versus anti-social economic behaviors[81]. For further example, left DLPFC has been linked to negative emotional responses to images[76], as well as children making incorrect choices to adult instructions[75]. Specifically in the Prisoner's Dilemma, the DLPFC has been implicated in the level of guilt felt after unreciprocated cooperation[82].

**Limitations**

Our results lend themselves to a nontrivial prediction: as the interaction distance increases, the more biased towards one's social tendency the response becomes. In the current design this means more cooperation, but given different incentives (i.e., where defection might be preferred by most participants) it predicts the opposite. In our sample, participants who on average defected more often than cooperated (23 out of 83) indeed cooperated significantly less with increasing distance but seemed relatively unaffected by group size (Fig. S4). This prediction should be examined more thoroughly in future studies (e.g., by manipulating the incentives within-person).

While our model fits the data quite well, we can observe some differences in fit quality across participants (Fig. 3E, F). This issue is likely the result of rich individual differences in PD strategies used by human participants[20], not all of which could be accurately captured by our model. Specifically, the model would struggle when faced with heuristic strategies which do not rely on reciprocity, such as GRIM or 'always defect'[20], or more sophisticated decision rules[83]. We did not find participants who used the GRIM strategy, but a small number did almost exclusively defect as noted in the exclusion criteria, and it is possible more complicated heuristics may be used beyond these. Indeed, certain personality characteristics have been previously associated with specific strategic choices[84]. Additionally, people are known to use theory of mind to form second-order beliefs and anticipate partners' actions[85,86]. Individual differences in personality and mentalizing depth provide excellent leads for future studies, which could provide more powerful predictions and understanding. One potential remedy for this in the context of our design is the development of mixture models[87], which can explicitly model the probability of a participant using one of a set of fixed choice strategies. Mixture models are notoriously difficult to fit due to identifiability and model-selection issues, so caution should be advised[88]. Our current model fit quality was high, and our main focus was on commonalities, not individual differences, so we did not pursue this avenue further.

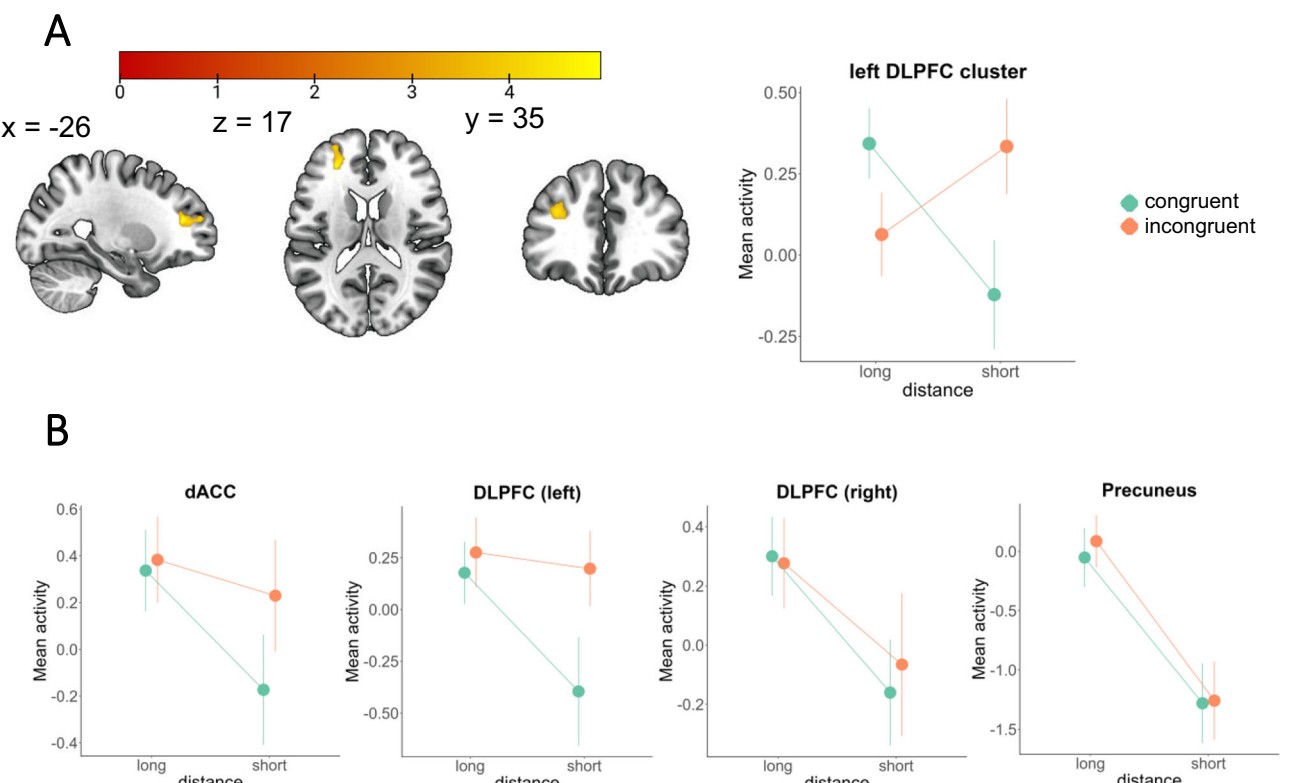

**Fig. 6 | Congruence by distance interaction effects. A** Whole-brain analysis showing a significant cluster within the left DLPFC $p < 0.05$ cluster-wise FWE corrected, cluster forming threshold $p < 0.001$. **B** Results from 4 *a priori* defined ROIs. Error bar length represents 2 standard errors around the mean.

Another limitation of our study is that it does not include an explicit measure of memory, which is only indirectly inferred from ID. This lends our findings to an alternative explanation, that it's the distance itself that leads to increased forgiveness[89]. While the current design cannot falsify this possibility, we find this explanation considerably less likely, due to the fact that it stands in direct opposition to the theoretical predictions[15], as well as evidence from empirical evidence indicating that memory capacity is a significant constraining factor for reciprocity[22,47]. Future studies could alleviate these concerns by either providing participants with unambiguous information of partner's previous choice or including an explicit memory test, while face-recognition and social memory fMRI localizers could help establish neural regions of interest.

Other popular explanations for increased cooperation include increased social pressure[55,90], and the role of preserving reputation[28,91]. These, however, can be discarded. Increased social pressure in our design is completely confounded by group size. Since including direct group effects in modeling did not improve model fit, the same is true for social pressure. Reputation-based explanation is not applicable, since all interactions were private, and avatar pictures were not re-used after group members left and later rejoined with new avatar faces. Thus, participants had no means of propagating reputation-based information, nor were they aware of partners' group sizes. While these factors did not influence behavior in our study, we believe they are important for understanding cooperation especially within more well-established social groups.

Finally, we also acknowledge the simplified nature of our task design compared to the real-world behaviors in groups. The current study examines the situations where basic forces of prosociality and reciprocal behavior operate. In the real world-setting, other forces such as reputation work to influence people's behavior. We hope this study will inspire future work, which could test whether and to what extent this mechanism generalizes to other group contexts, such as larger, or more established groups, where group dynamics are affected by culture, hierarchy, personality traits, and a long history of prior interactions.

## Conclusion

Our findings provide insight into understanding how cooperation can emerge between members of newly forming and dynamically maintained groups, linking shorter-term changes in group size with research related to much longer temporal cycles (e.g. individual development or evolution). We propose that group growth creates an increase demand for social memory storage and retrieval, which, in the short term, leads to strategical adjustments towards one's default social tendency, while in the long term (brain maturation, evolutionary changes) induces slower changes in processing capacity or prosocial tendencies, an important direction for future work.

Together, the proposed neuro-cognitive mechanism can explain how groups can sustain (or even increase) cooperative interactions, despite diminishing economic incentives to do so, as the group gets larger. Given favorable initial conditions, such as social tendency to cooperate and an economic structure that does not overly punish cooperation, these cognitive processes can lead to the emergence of large-scale cooperative groups and societies, which define the success of human civilization[92].

## Data availability
The full behavioral data, fMRI data allowing for figure generation, and a video demonstration of the task are freely available at https://osf.io/45r3x https://doi.org/10.17605/OSF.IO/45R3X.

## Code availability
The code for all statistical analyses in the paper is freely available at https://osf.io/45r3x https://doi.org/10.17605/OSF.IO/45R3X.

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

## Acknowledgements

The RIKEN Center for Brain Science (CBS)—Toyota Collaboration Center (BTCC) (LP3009219) funded W.Z., R.P.B. and R.A. R.P.B. was additionally supported by the RIKEN Special Postdoctoral Researcher Program. The funders had no role in study design, data collection and analysis, decision to publish or preparation of the manuscript.

## Author contributions

M.H. and R.A. planned and implemented the behavioral experiments. W.Z, R.P.B., M.H., and R.A. all analyzed the data and contributed to the manuscript writing.

## Competing interests

The authors declare no competing interests.
