## [Peer Review file · Communications Psychology]

A Neurocognitive Mechanism for Increased Cooperation During Group Formation

Corresponding Author: Dr Wojciech Zajkowski

Version 0:

Decision Letter:

Dear Dr Zajkowski,

Thank you for your patience during the peer-review process. Your manuscript titled "Forget and Forgive: A Neurocognitive Mechanism for Increased Cooperation During Group Formation" has now been seen by 3 reviewers, and I include their comments at the end of this message. They find your work of interest but raised some important points. We are interested in the possibility of publishing your study in Communications Psychology, but would like to consider your responses to these concerns and assess a revised manuscript before we make a final decision on publication.

We therefore invite you to revise and resubmit your manuscript, along with a point-by-point response to the reviewers. Please highlight all changes in the manuscript text file.

Editorially, we consider that you first address some conceptual clarifications. R1 questioned the rationale for selecting the task, and both R1 and R3 raised the potential discrepancy between the model's computer-based application and real-world group dynamics. R2 also requested you address the (causal) relationship between group size and cooperation rates. Note that we cannot permit directional claims on the basis of correlational data.

In sum, we ask that you provide a strong rationale for your choices, clearly acknowledge limitations (such as untestable directionality claims, confounds, and artificiality or other constraints in the task). We require that you discuss these in the limitations section. Moreover, please also revise the overall presentation so that the account given in all parts of the text, including Title, Abstract, and Introduction remains close to the data. While alternative accounts should be noted, please be mindful to avoid speculation.

Second, please conduct a complementary analysis and add more clarifications on LLM and computational modeling as requested by both R1 and R2.

Finally, as emphasized in our guidelines (attached), please provide additional information about which hypotheses and methods were pre-specified versus ad hoc, including regarding the choice of ROIs.

I am attaching an Editorial Requests Table that details critical reporting requirements for the revised manuscript. Please attend to each item and ensure your manuscript is fully compliant. We are requesting that your manuscript aligns with these requirements as this facilitates the evaluation of your manuscript, reducing delays in re-review and potential future acceptance. If your revised manuscript is not aligned with these requests on major issues, such as those concerning statistics, it may be returned to you for further revisions without re-review. Additional information can be found in our style and formatting guide https://www.nature.com/documents/commspsychol-style-formatting-guide-accept.pdf Communications Psychology formatting guide.

Please use the following link to submit your
- revised manuscript,

- point-by-point response to the referees' comments,
- cover letter (as a separate document),
- the Editorial Policy Checklist (see below),
- the Reporting Summary (see below), and
- the completed Editorial Request Table (attached):

Link Redacted

Best regards,

Yafeng Pan

Yafeng Pan, PhD
Editorial Board Member
Communications Psychology
orcid.org/0000-0002-5633-8313

REVIEWER EXPERTISE:

Reviewer #1: Cooperation/fMRI

Reviewer #2: Cooperation/fMRI/Computational modeling

Reviewer #3: Cooperation/fMRI

REVIEWER REPORTS:

Reviewer #1 (Remarks to the Author):

The study is very interesting, complex and very well written. It investigates the impact of group size on cooperation through a dynamic, network-based prisoner's dilemma experiment coupled with fMRI analysis. While the participants demonstrated increased cooperation as group sizes expanded, the study uncovered that the sheer increase in group size did not directly foster cooperation. Instead, cooperation was shaped by a combination of participants' inherent prosocial tendencies and dynamic reciprocal strategies, influenced by their confidence in memory. This interaction increasingly favored prosocial tendencies as the memory demands escalated in larger groups. Memory confidence correlated with brain activity in the fusiform gyrus and precuneus, while its integration with prosocial tendencies manifested in the left DLPFC and dACC. The study concludes that the interplay between uncertainty in memory recall during reciprocal interactions (i.e., forgetting) and individual prosocial preferences is essential for cooperation in larger, dynamically forming groups.

I have a several comments that I hope authors can address.

The first two point raises a conceptual question.

- I'm curious why the author chose to use the prisoner's dilemma (PD) instead of repeated trust games for the study. The inherent nature of the PD suggests that the optimal strategy is to defect. In the text, the authors explain that a reputation-based rationale does not apply here because all interactions were private, and avatars were not reused once group members departed and then rejoined with new identities. Consequently, participants had no way to disseminate reputation-based information or to be aware of their partners' group sizes. Therefore, selecting the PD game for testing cooperation is pivotal, as it does not establish trust or entail consequences for the participants' actions. This could potentially increase the likelihood of free-riding behavior, as participants might focus solely on maximizing their personal gains.

- The second point addresses the authors' emphasis on the significance of dyadic interactions, which are prevalent in our daily lives and play a crucial role in the functioning of small groups such as research teams, sports teams, corporate teams, and music bands. However, the authors later mention that the models developed in this research are intended to be applied to members of newly forming and dynamically maintained groups. This raises a question: Can these models accurately

predict behaviors in the previously mentioned groups, like sports teams, where there are strong social interactions and a well-established level of trust within the group? Maybe authors may address the potential discrepancy between the model's application and real-world group dynamics.

- The authors did not discuss the potential impact of personality traits on cooperation in the Prisoner's Dilemma (PD). Personality traits could influence the likelihood of making effective choices or the tendency to engage in free-riding behavior. For example, research by Malesza (2020) explores how Dark Triad traits affect decisions in the PD game, suggesting that certain personality characteristics can significantly alter cooperative behaviors. Additionally, Guzmán et al. (2020) integrate personality traits into a game-theoretic model of reciprocity and trust, further underscoring the role these traits play in shaping interactions within such frameworks. These examples highlight a gap in the current research that could be vital for understanding the complexities of cooperation in the PD context.

References:

- Malesza, M. (2020). The effects of the Dark Triad traits in prisoner's dilemma game. *Current Psychology*, 39, 1055–1062. <https://doi.org/10.1007/s12144-018-9823-9>

- Guzmán, R., et al. (2020). A game-theoretic model of reciprocity and trust that incorporates personality traits. *Journal of Behavioral and Experimental Economics*, 84, 101497.

- Another factor noted in the limitations section by the authors is memory capacity. The approach to associate the number of trials in which a participant reciprocated with an opponent purely with memory and its decay might be overly mechanistic. It appears that neither memory capacity nor face familiarity was directly measured in the study. This raises the question of whether some faces were inherently more memorable than others, and whether participants actively paid attention to faces to remember their opponents. For instance, in a large group such as an orchestra with around 40 members, what would the dynamics be in cooperation if members actually remembered each other? Additionally, memory is intricately linked to emotions; the betrayal by someone might be perceived more personally by one individual than another, leading to stronger memory associations. These emotional factors could significantly influence memory recall and, consequently, the dynamics of cooperation and reciprocity within the group.

-The authors stated- Specifically, we assume that the memory fidelity of the current partner's previous choice decays exponentially as a function of ID. By reducing the value of reciprocation, the relative social tendency now dominates the choice function, making the prediction that worse memory should lead to an increase in the individual default tendency (either cooperation or defection, depending on the subject). What do you mean by depending on subject?

- The rationale for conducting a post hoc analysis of the ventromedial prefrontal cortex (VMPFC) isn't very clear. While the general linear model (GLM) highlighted significant results in the nucleus accumbens (nACC), a region well-known for its association with value processing, the decision to additionally examine the VMPFC—merely because it is also linked to value—raises questions. Other regions like the orbitofrontal cortex (OFC), putamen, and caudate are similarly connected to value but were not singled out for similar analysis. Moreover, Neurosynth identified a substantial cluster in the VMPFC. It remains unclear whether this was the primary cluster peak or if its inclusion was based on its size and relevance. This clarification could provide deeper insight into the selection criteria and the broader implications of these findings in the study's neurological assessments.

-Could elaborate more on role for finding of association of DLPFC and betrayal?

-Could the authors provide more detail in the discussion on how the model might function or be integrated within well-established social groups? Additionally, it would be beneficial to explore the potential role of dark personality traits and reputation levels in predicting behavior. This deeper analysis could enhance understanding of the model's applicability to diverse social dynamics and offer insights into how individual differences in personality and perceived reputation might influence group interactions and outcomes.

Minor points:

In the Table 1 would be nice to add for readers descriptions of M models

Why 4-mm smoothing was used? Normally 8, 6 or even 10, but 4?

Reviewer #2 (Remarks to the Author):

1. Throughout the manuscript, the authors argued that group size increases cooperation. However, it is unclear whether the current design and results support the conclusion. On the design, it is possible that higher cooperation tendency maintains a larger group size, rather than the way around. That is, how many partners a participant had depended on how cooperative they were. On the results, the current study revealed that larger distance-related forgetting resulted in choices based on intrinsic social tendency instead of value associated with group size. Together, it is likely that larger group size one

- maintains is just a byproduct (not a reason) of one's higher cooperation rates.
2. The exclusion of 4 participants who defected in most trials should be justified further.
 3. For the LLM analyses, why effects of GS and ID were estimated in separated models, given that those effects were estimated simultaneously in the computational models?
 4. For the computational modeling, the authors only considered value of cooperation in their valuation functions, so that the value of defection was fixed to 0. However, there are several other factors that may impact participants' choices: (1) the monetary reward. It is likely that participants made their choices based on the tradeoff between social and monetary reward, but the current models did not capture the tradeoff; (2) the second-order belief of how their own choices may impact partner's follow-up decisions, see also Hampton et al. (2008) and Hill et al. (2017).
 5. Please provide relevant references (e.g., theoretical backgrounds, applications, and/or simulations) to justify the application of beta-restricted priors.
 6. How the ranges of parameters were set? Please provide justification. What the range of tau (inverse temperature)?
 7. VRC were fixed to 1, please provide justification. The identifiability issue may be due to the limited number of data or the problem of models. The authors may provide more detailed descriptions about the issue.
 8. For the GLM3, would value of forgiveness and value of betrayal be highly correlated with each other?
 9. Many ROIs were constructed, but the authors did not provided relevant hypotheses on why specific ROIs were chosen.
 10. Some typos in the manuscript:
Abstract: `...one's individual prosocial preference in a core pillar' should be 'is a core pillar'
Discussion: `...and plays a major role is brain state transitions' should be 'in brain state transitions'.

Reviewer #3 (Remarks to the Author):

This article describes a novel paradigm described as assessing the role of time and forgetting in cooperation in increasingly large groups. The overall conceptualization and implementation of the study is novel and likely to be of wide general interest.

I would also like to say that the main text of the article is very well written... the authors should be commended on crafting a manuscript that was such a pleasure to read.

I do have a few concerns about the methods and the conclusions that can be drawn from the results, as well as some thoughts about the framing of the article, which I describe in more detail below:

Introduction:

- Overly specific lists of regions and their function on p6--seems to suggest that, e.g., DLPFC plays some unique and specific role in betrayal. Does it?

Methods/Results

- On p12 of the Discussion the authors state: "our model predicts that the memory trace of the current partner's previous choice decays with time." Unless there was a design feature that I missed, time is confounded with the number of prior interactions in this paradigm. In theory, different memory processes would be associated with forgetting depending on the cause. Can the authors distinguish whether the cause of forgetting was the passage of time (WM) versus the number of intervening interactions (interference). I agree with the authors' statement in the Limitations that a lack of any explicit tests of memory phenomenon are a serious limitation of a study that makes strong claims, starting in the title, that the observed effect reflects specific memory mechanisms.

- The Methods states that the avatar faces were taken from Ekman face set neutral faces, but the images in Fig 1 are from the NimStim set. Which is correct?? In either case, I wonder why that set was used. Usually that set is selected for its emotional faces. Were there other conditions in which the emotional faces were used that are not being reported?

- I note that the study is not preregistered and wonder why, and whether the authors can provide additional information about which hypotheses and methods were pre-specified versus ad hoc. (For example, I found it surprising that there was no evidence of hippocampus playing a role in forgetting or episodic memory during the task, and wondered if that was really what was predicted.)

- Also, these are somewhat puzzling faces to pick. In either case (Ekman or NIMSTIM) they are of Americans and it seems like this study was run in Japan. Some explanation for this choice of stimuli is required, and at least a brief discussion of the limitations it creates. One big one is of course that Americans (who can often be recognized as such regardless of race from their facial behavior) are 'outgroup' members for people in Japan--at least, they would be described as outgroups by default in many studies. And in general outgroup members elicit less trust and thus cooperation, particularly in more collectivist societies like Japan. So this study design seems like it very likely biased behavior typical to baseline. Do the authors have any evidence to confirm that the choice of these avatar faces did not change participant behavior relative to faces of Japanese students or non-outgroup faces (cartoon characters, abstract images)

- Why was age not recorded for 1/3 of participants?

- I'm confused by the line: "The partner was given the same stay/switch decision and was more likely to switch if the subject was more defective recently." I thought the partners were "prescheduled computer programs"? How exactly was this decision programmed in? It seems like a lot of the study is riding on the precise "behavior" and "decisions" of the computer that is not entirely clearly specified.

- I'm less sure that this study is capturing actual group dynamics if people only ever interact with computers. It seems like human behavior might be shaped by a context like this in a way that would meaningfully change prosocial responding, which can't be tested when each human responder is only interacting with computers, especially if all "partners" were

programmed with same algorithm. A key feature of real groups, and the decision to cooperate in sequential prisoner's dilemmas, is variation.

- I'm also not sure if I'd call this a "group" in any meaningful way. As far as I understand it, each person is only aware of their own ongoing interaction with each avatar, but has no idea if they are interacting with each other or not, or have any other kind of collective identity, which would be intrinsic to it being a "group."

- I had some concerns upon reading the line: "The subjects overall seemed exhausted but appeared to enjoy the experience in the task by the end." How many trials were in the task that rendered participants 'exhausted'? I searched for the length of the study or number of trials and could not locate that information. Is there a concern that this exhaustion is part of the 'forgetting' that might be independent of the specific proposed memory-based mechanisms?

- As far as the neural substrates go, can the authors explain why the DLPFC would have a special role in responding to betrayal given this region's involvement in "forgiveness" as well? Hard to say for sure from the images, but the activation patterns appear to be overlapping. Could this region be doing something more general related to reciprocity? The functional connectivity analysis seems potentially consistent with that. (A contrast between responses to forgiveness versus betrayal would seem to be helpful here)

EDITORIAL POLICIES

We ask that you ensure your manuscript complies with our editorial policies and reporting requirements.

To that end, we require revised manuscripts to be accompanied by two completed items: a reporting summary that collects information on study design and procedure, and an editorial policy checklist that verifies compliance with all required editorial policies.

- <https://www.nature.com/documents/nr-reporting-summary.zip> Nature Research Reporting Summary
- <https://www.nature.com/documents/nr-editorial-policy-checklist.pdf> Editorial Policy Checklist

All points on the policy checklist must be addressed. Your revised manuscript can only be sent back to the referees if these checklists are completed and uploaded with the revision.

Notes: If you have submitted a Stage 1 Registered Report, Review, Primer, Comment, or Perspective you do not need to submit these forms. If you have already submitted these forms, you may disregard this request.

Communications Psychology is committed to improving transparency in authorship. As part of our efforts in this direction, we are now requesting that all authors identified as 'corresponding author' create and link their Open Researcher and Contributor Identifier (ORCID) with their account on the Manuscript Tracking System prior to acceptance. ORCID helps the scientific community achieve unambiguous attribution of all scholarly contributions. You can create and link your ORCID from the home page of the Manuscript Tracking System by clicking on 'Modify my Springer Nature account' and following the instructions in the link below. Please also inform all co-authors that they can add their ORCID to their accounts and that they must do so prior to acceptance.

Version 1:

Decision Letter:

** Please ensure you delete the link to your author homepage in this e-mail if you wish to forward it to your coauthors

** Dear Dr Zajkowski,

Your manuscript titled "Forget and Forgive: A Neurocognitive Mechanism for Increased Cooperation During Group Formation" has now been seen by two of the reviewers, whose comments appear below. In light of their advice I am delighted to say that we are happy, in principle, to publish a suitably revised version in Communications Psychology.

We therefore invite you to revise your paper one last time to address the remaining concerns of our reviewers and a list of editorial requests. At the same time we ask that you edit your manuscript to comply with our format requirements and to maximise the accessibility and therefore the impact of your work.

EDITORIAL REQUESTS:

SUBMISSION INFORMATION:

OPEN ACCESS:

* DATA AVAILABILITY:

<https://mts-commspsychol.nature.com/cqi-bin/main.plex>

Best regards,

Marike Schiffer

on behalf of
Yafeng Pan, PhD
Editorial Board Member
Communications Psychology

Marike Schiffer, PhD
Chief Editor
Communications Psychology

REVIEWERS' COMMENTS:

Reviewer #1 (Remarks to the Author):

Authors addressed all my comments. Thank you.

Reviewer #2 (Remarks to the Author):

The authors have addressed all of my concerns.

Response to Reviewer Comments

In our response, we numbered the questions, according to reviewer number and question number (e.g., 2.4. refers to the 4th comment of reviewer 2). Below we address all the comments and questions. We provide responses directly below each point. Reviewer comments are written in italic, while our responses are in plain font style. Additionally, we use a different font (Times New Roman) when citing excerpts from the new version of the manuscript.

Reviewer #1 (Remarks to the Author):

The study is very interesting, complex and very well written. It investigates the impact of group size on cooperation through a dynamic, network-based prisoner's dilemma experiment coupled with fMRI analysis. While the participants demonstrated increased cooperation as group sizes expanded, the study uncovered that the sheer increase in group size did not directly foster cooperation. Instead, cooperation was shaped by a combination of participants' inherent prosocial tendencies and dynamic reciprocal strategies, influenced by their confidence in memory. This interaction increasingly favored prosocial tendencies as the memory demands escalated in larger groups. Memory confidence correlated with brain activity in the fusiform gyrus and precuneus, while its integration with prosocial tendencies manifested in the left DLPFC and dACC. The study concludes that the interplay between uncertainty in memory recall during reciprocal interactions (i.e., forgetting) and individual prosocial preferences is essential for cooperation in larger, dynamically forming groups.

I have a several comments that I hope authors can address.

The first two point raises a conceptual question.

1.1.

- I'm curious why the author chose to use the prisoner's dilemma (PD) instead of repeated trust games for the study. The inherent nature of the PD suggests that the optimal strategy is to defect. In the text, the authors explain that a reputation-based rationale does not apply here because all interactions were private, and avatars were not reused once group members departed and then rejoined with new identities. Consequently, participants had no way to disseminate reputation-based information or to be aware of their partners' group sizes. Therefore, selecting the PD game for testing cooperation is pivotal, as it does not establish trust or entail consequences for the participants' actions. This could potentially increase the likelihood of free-riding behavior, as participants might focus solely on maximizing their personal gains.

We appreciate the reviewers' inquiry as to whether repeated trust game in the network setting can give us additional insights. Different types of trust or mutual exchange games (such as Prisoner's Dilemma, Trust Game, or Ultimatum Game) carry their unique sets of advantages and disadvantages. This comment hit the proverbial nail on the head, by emphasising the '**accountability**' aspect of the design: interactions were repeated, and the avatars were unique, minimising the incentive to exploit the game without facing future

consequences. An important caveat here being that 'free-riding' behavior is impossible in this specific task variant, since choices are made simultaneously, and within dyads. For PD, free-riding traditionally refers to an effect in simultaneous play with larger groups all making their choice at once rather than dyadically:

Gracia-Lázaro, Carlos, et al. "Human behavior in Prisoner's Dilemma experiments suppresses network reciprocity." *Scientific reports* 2.1 (2012): 325.

One favourable aspect of PD compared to some of the other trust games is that the interactions are symmetrical, eliminating any potential confounding effects associated with asymmetry (e.g. in UG, one of the players is the *proposer*, while the other is the *receiver*, while in TG: *trustor* and *trustee*). These dynamics make TG and UG more suitable for studying non-simultaneous, sequential choices, which enable free-loading behavior. Furthermore, the dyadic network-based PD game gives us advantages in terms of more dynamic behavior with quick updating of the network structure by the simultaneous decisions of cooperation/defections and the need for single partner tracking. Based on this, we think our variant of PD lends itself better, as an experimental protocol, for testing the specific hypotheses of early group formation.

1.2

- The second point addresses the authors' emphasis on the significance of dyadic interactions, which are prevalent in our daily lives and play a crucial role in the functioning of small groups such as research teams, sports teams, corporate teams, and music bands. However, the authors later mention that the models developed in this research are intended to be applied to members of newly forming and dynamically maintained groups. This raises a question: Can these models accurately predict behaviors in the previously mentioned groups, like sports teams, where there are strong social interactions and a well-established level of trust within the group? Maybe authors may address the potential discrepancy between the model's application and real-world group dynamics.

We appreciate the reviewer's comment raising the important point of distinctions between newly forming groups and well-established ones. We acknowledge that there are substantial differences between these two types of groups. The former is concerned with the relatively shorter temporal scope of past behaviors, whereas the latter is concerned with the norms established through the long-term history of interactions. In addition, the relative anonymity of the current design might resemble the encounters in online social interactions, such as those in social networking services.

More generally, we also acknowledge the simplified nature of our task design, as compared to real-world group interactions (a common issue in neuroscience research where tasks must be simplified abstractions of complex real-world behavior to make fMRI analysis feasible). Nevertheless, our current task design is informative of the general mechanisms driving social interactions in newly forming groups. We believe that the mechanism described in the paper can be applied to many circumstances of newly forming groups in which the history of recent interactions are important but cannot be easily recalled.

Lastly, we intend for this work to aid in encouraging and inciting many new future directions of economic and psychological group dynamics in the laboratory context, and thus believe this is an important comment to guide future work. For example, it would be interesting in the future to conduct the experiments of network-based PD games with such acquainted members, e.g. schoolmates or family members. We've further clarified this point in the discussion in the sentence:

Lines 518-525:

Finally, we also acknowledge the simplified nature of our task design compared to the real-world behaviors in groups. The current study examines the situations where basic forces of prosociality and reciprocal behaviors operate. In the real world-setting, other forces such as reputation work to influence people's behavior, especially in the established groups. In contrast, our work focus on the forces of prosociality and reciprocations in the context of early group formation. We hope this study will inspire future work, which could test whether and to what extent this mechanism generalizes to other group contexts, such as larger, or more established groups, where group dynamics are affected by culture, hierarchy, personality traits, and a long history of prior interactions.

1.3

- The authors did not discuss the potential impact of personality traits on cooperation in the Prisoner's Dilemma (PD). Personality traits could influence the likelihood of making effective choices or the tendency to engage in free-riding behavior. For example, research by Malesza (2020) explores how Dark Triad traits affect decisions in the PD game, suggesting that certain personality characteristics can significantly alter cooperative behaviors. Additionally, Guzmán et al. (2020) integrate personality traits into a game-theoretic model of reciprocity and trust, further underscoring the role these traits play in shaping interactions within such frameworks. These examples highlight a gap in the current research that could be vital for understanding the complexities of cooperation in the PD context.

References:

- Malesza, M. (2020). The effects of the Dark Triad traits in prisoner's dilemma game. Current Psychology, 39, 1055–1062. <https://doi.org/10.1007/s12144-018-9823-9>*
- Guzmán, R., et al. (2020). A game-theoretic model of reciprocity and trust that incorporates personality traits. Journal of Behavioral and Experimental Economics, 84, 101497.*

Yes, we believe individual differences can have a large impact on strategy choice and behavior and think that this is an important future direction of research. We acknowledge this in the paragraph starting on line 477 of the manuscript. We have now expanded this paragraph, including provided references (Refs 68 & 69), which now reads as follows:

Lines 477-495:

While our model fits the data quite well, we can observe some differences in fit quality across participants (Fig. 3EF). This issue is likely the result of rich individual differences in PD strategies used by human participants²⁰, not all of which could be accurately captured by our model. Specifically, the model would struggle when faced with heuristic strategies which do not rely on reciprocity, such as GRIM or ‘always defect’²⁰, or more sophisticated decision rules⁶⁸. We did not find participants who used the GRIM strategy, but a small number did almost exclusively defect as noted in the exclusion criteria, and it is possible more complicated heuristics may be used beyond these. Indeed, certain personality characteristics have been previously associated with specific strategic choices⁶⁹. Additionally, people are known to use theory of mind to form second-order beliefs and anticipate partners’ actions^{70,71}. Individual differences in personality and mentalizing depth provide excellent leads for future studies, which could provide more powerful predictions and understanding. One potential remedy for this in the context of our design is the development of mixture models⁷², which can explicitly model the probability of a participant using one of a set of fixed choice strategies. Mixture models are notoriously difficult to fit due to identifiability and model-selection issues, so caution should be advised⁷³. Our current model fit quality was high and our main focus was on commonalities, not individual differences, so we did not pursue this avenue further.

Also see our comment above about why free riding would not occur in our specific dyadic network PD task variant.

1.4

- Another factor noted in the limitations section by the authors is memory capacity. The approach to associate the number of trials in which a participant reciprocated with an opponent purely with memory and its decay might be overly mechanistic. It appears that neither memory capacity nor face familiarity was directly measured in the study. This raises the question of whether some faces were inherently more memorable than others, and whether participants actively paid attention to faces to remember their opponents. For instance, in a large group such as an orchestra with around 40 members, what would the dynamics be in cooperation if members actually remembered each other? Additionally, memory is intricately linked to emotions; the betrayal by someone might be perceived more personally by one individual than another, leading to stronger memory associations. These emotional factors could significantly influence memory recall and, consequently, the dynamics of cooperation and reciprocity within the group.

This is another excellent point. We acknowledge that the simplicity of the mechanistic explanation proposed is not sufficient to fully explain all the intricacies driving cooperative behaviour, but believe it points to a more general process. This a clear trade-off between trying to explain the most invariant and general mechanism that can capture behavioral

outcomes in a wide variety of circumstances (individual differences, specific stimuli, etc), and diving deeper into the weeds of accurately describing the plethora of contextual factors driving individual choice, and our study falls closer to the former end of this spectrum.

First please note that, as shown in Figure 1, the faces of all current group members are shown in the bottom of the task screen, while the current partner is above them. Thus, the subjects do not have to remember the faces of current group members, just the recent interactions with each face which we feel it's reasonable to assume humans can readily do on the timescale of seconds to minutes at least, though of course with decreasing accuracy as both the number of partners and distance to past interactions increases. This last effect is what we were trying to model and understand.

We added the full stimulus set as a supplementary figure:

Figure S1. All 30 neutral faces used in the experiment. Each participant interacted with the same set of partner faces, introduced in random order. All stimuli were chosen from the neutral set of faces from the NimStim set of facial expressions (Tottenham et al., 2009)

Further, addressing the two specific issues mentioned in the above comment:

1. We agree that a defection can potentially leave a stronger mark on one's memory, and in the initial exploratory phase of the modelling procedure, we did test out models which had separate memory rates for defection and cooperation. These models however did not fit well to the data and this idea was abandoned at an early modelling stage.

2. As pointed out, different faces might induce differences in behavior. For example, some might be perceived as more or less trustworthy, friendly or memorable. To test this, we performed a simple analysis, calculating the average cooperation and reciprocity rate *per* face. Note that each new face was randomly drawn from a pool of 30 possible faces when a

new partner joined a group, and faces that left a group were never re-used within a session. The cooperation rates per face range between 0.459 (the least 'trustworthy' face) and 0.677 (the most 'trustworthy' face), with mean of 0.583 and a SD of 0.058. Thus, in a first check, there were not major differences in cooperation rate per face. We explore more below.

For reciprocity, the range across faces is between 0.625 and 0.757, with the mean off 0.699 and SD of 0.033. This suggests that even if differences in face identity did influence behavior to some small degree, there were no 'extreme' faces, which would induce very high or very low levels of cooperation or reciprocity.

Importantly, both of these quick analyses do not account for random sampling differences in cooperation rate of the partner across faces. To test this, we performed a regression analysis on pooled data, predicting reciprocity from individual faces. The model can be expressed symbolically as:

reciprocity ~ faceID

Here, faceID is a categorical factor variable with 30 different categories, corresponding to partners' faces. Significant effects would indicate that a given face predicts reciprocity meaningfully better than the average.

The analysis did not provide significant coefficients for any of the 30 faces. The largest positive coefficient was equal to 0.078 ($t=1.716$, $p=0.086$) for face 4. The largest negative coefficient was equal to -0.046 ($t=-1.031$, $p=0.302$) for face number 18. These results strongly suggest that face identity was not a significant predictor of reciprocity.

We added these analyses, together with tests for effects of Gender and Race as well as qualitative plots, to the Supplementary Materials (section: *Effects of Partner Face*). Race and gender of faces were not found to influence the cooperation rate of this Japanese subject pool.

1.5

-The authors stated- Specifically, we assume that the memory fidelity of the current partner's previous choice decays exponentially as a function of ID. By reducing the value of reciprocation, the relative social tendency now dominates the choice function, making the prediction that worse memory should lead to an increase in the individual default tendency (either cooperation or defection, depending on the subject). What do you mean by depending on subject?

We mean that the default cooperative tendencies of the subjects are different. There were two types of participants: those who predominantly cooperated (majority, 62/83 participants), and those who predominantly defected (21/83). For the first group, the default social tendency is cooperation, but for the second, it's defection. Hence, increase in the default social tendency is opposite in these two groups: it leads to higher cooperation within cooperators, and higher defection among defectors. Additionally, the cooperative and defective subjects differ in terms of the strength of the default bias. A detailed analysis of this

pattern is presented in the Supplementary Materials in the section titled: Behavioral patterns for participants with negative social tendency.

1.6

- *The rationale for conducting a post hoc analysis of the ventromedial prefrontal cortex (VMPFC) isn't very clear. While the general linear model (GLM) highlighted significant results in the nucleus accumbens (nACC), a region well-known for its association with value processing, the decision to additionally examine the VMPFC—merely because it is also linked to value—raises questions. Other regions like the orbitofrontal cortex (OFC), putamen, and caudate are similarly connected to value but were not singled out for similar analysis. Moreover, Neurosynth identified a substantial cluster in the VMPFC. It remains unclear whether this was the primary cluster peak or if its inclusion was based on its size and relevance. This clarification could provide deeper insight into the selection criteria and the broader implications of these findings in the study's neurological assessments.*

While it is true that areas such as OFC, putamen and caudate have been linked to value in different contexts, it is the VMPFC that is often considered the 'root of all value', or the neural common currency of choice (see: reference 31). This hypothesis has been also confirmed in relation to social value specifically, concluding that VMPFC integrates positive and negative value signals, reflecting subjective evaluation (reference 90). Therefore, the choice of the VMPFC cluster was an 'a priori', theory-driven choice, and was influenced by the size of the cluster in the *Neurosynth* meta-analysis.

This information, together with the motivation for choosing specific ROIs has been added to the ROI Analysis section of Methods:

ROI analysis

All ROIs were independently defined based on an online meta-analysis using keywords in the *Neurosynth*³² database. The keywords were dictated by the hypothesized function of these areas within the context of our task. In order to obtain a reasonable sample size of studies, the keywords were as general as possible. Additionally, the activity clusters needed to be located within the anatomical bounds of the region for which the ROI was being obtained. All ROIs were spherical, with varying volumes. The ROIs were dictated by a) anatomical size of the structure of interest, and b) the size of the functional cluster, within the arbitrary limits of not being smaller than 5 mm³, and not larger than 10 mm³. ROI analysis was performed by extracting whitened and filtered voxels within an ROI from a contrast of interest and averaging the voxel values within participants. The vector of averaged beta weights was then compared against 0 using a t-test.

Value of cooperation. The VMPFC ROI was defined as 10 mm³ sphere, centered at the peak activity for the keyword 'value' (peak MNI coordinates: x=0, y=40, z=-8; 407 studies included in the database). The choice of VMPFC was driven by the *a priori* expectation of

the region being involved in subjective value processing³¹ (including social contexts⁹⁰).

1.7.

-Could elaborate more on role for finding of association of DLPFC and betrayal?

For interpreting the connection between the DLPFC and betrayal, we've focused on both (1) multi-timescale computational role of the DLPFC and (2) the connection in prior literature to the DLPFC in terms of betrayal-relevant concepts such as emotions and guilt, antisocial behavior, and executive control.

For (1), we've referenced and cited related literature extensively so will not expand unless further asked, see for example:

Lines 445-461:

DLPFC can represent information on a wide range of timescales, from single to multiple trials back in the session history^{39,58}. As such, it is well suited for functions related to the preservation of behavioral consistency, such as maintaining intentions⁵⁹, as well as dynamic control, like reappraisal⁶⁰ or cognitive resource allocation⁶¹. In social game settings, DLPFC has been associated with exerting cognitive control in service of goal-directed behavior, necessary in situations of high cognitive load^{60,62,63}. Overall, the ability to integrate information from different temporal scales is crucial for computing value signals, where the value of reciprocation is moderated by the ID-dependent context.

Activity within the left DLPFC and dACC reflects not only the context itself (current partner's previous choice), but also its reliability, making the signal sensitive to all three crucial variables driving choice (social tendency, value of reciprocation, and memory retention). Interestingly, both of those areas were previously implicated with exerting cognitive control *via* the inhibition of tempting responses^{64,65}. Our results suggest that their role might be more general, reflecting the temporal integration of value-relevant signals representing conflicting behavioral tendencies.

For (2), the connection to DLPFC with betrayal-related neural signals we've expanded the references and discussion with this new content:

Lines 462-468:

Previous literature has also linked the DLPFC to functions that are conceptually related to betrayal, such as cognitive control⁶⁰ and modulation of social versus anti-social economic behaviors⁶⁶. For further example, left DLPFC has been linked to negative emotional responses to images⁶¹, as well as children making incorrect choices to adult instructions⁶⁰. Specifically in the Prisoner's Dilemma, the DLPFC has been implicated in the level of guilt felt after unreciprocated cooperation⁶⁷.

1.8

-Could the authors provide more detail in the discussion on how the model might function or be integrated within well-established social groups? Additionally, it would be beneficial to explore the potential role of dark personality traits and reputation levels in predicting behavior. This deeper analysis could enhance understanding of the model's applicability to diverse social dynamics and offer insights into how individual differences in personality and perceived reputation might influence group interactions and outcomes.

The current study examines a controlled laboratory situation where basic forces of prosociality and reciprocal behavior operate. In the real world-setting, other forces such as short- and long-term reputation can also influence people's behaviors in more complex ways. We point to this, as well as associated studies in the *Discussion*:

Lines 508-518:

Other popular explanations for increased cooperation include increased social pressure^{29,75}, and the role of preserving reputation^{28,76}. These, however, can be discarded. Increased social pressure in our design is completely confounded by group size. Since including direct group effects in modelling did not improve model fit, the same is true for social pressure.

Reputation-based explanation is not applicable, since all interactions were private, and avatar pictures were not re-used after group members left and later rejoined with new avatar faces. Thus, participants had no means of propagating reputation-based information, nor were they aware of partners' group sizes. While these factors did not influence behavior in our study, we believe they are important for understanding cooperation especially within more well-established social groups.

The factor of personality is also interesting to consider. Indeed, we have observed trait-like effects of the default prosocial tendency in our data set. People's default tendency differs in its intensity and its direction. This stable tendency, which is not affected by recent experiences, can be interpreted partly as some personality/trait-like characteristics, and could be studied and correlated with trait survey results in future work. Alternatively, this

stable tendency may be modulated by long-term experiences and cultural factors as well. For example, it would be worthwhile in future work to compare whether our current Japanese participants display different baseline tendencies compared to other cultures, and how personality measures interact with cultural effects.

We have acknowledged all these points in the discussion section, between lines 478-496 and 508-518.

Minor points:

In the Table 1 would be nice to add for readers descriptions of M models

Thank you for the suggestion, it has been implemented.

Lines 274-277:

Table 1. LOOIC and WAIC model scores. M1: no memory, no group effect; M2: perfect memory, no group effect; M3: imperfect memory (forgetting), no group effect; M4: no memory, group effect; M5: perfect memory, group effect; M6: imperfect memory (forgetting), group effect.

Why 4-mm smoothing was used? Normally 8, 6 or even 10, but 4?

We used the rule-of-thumb recommendation provided in SPM documentation of using a smoothing kernel twice the size of a single voxel dimension (2x2x2). A 4-mm kernel provides higher spatial resolution than 7 or 8mm ones. Of course, there is a tradeoff, since in case of bad quality data (e.g. a lot of movement artifacts) small smoothing kernels could lead to 'jagged', discontinuous activation patterns. Since movement artifacts in our dataset were minimal, we believe this choice was justified.

Also see prior relevant work using this kernel size:

“ Although 4 mm is only slightly larger than the voxel size, we chose this value because previous research suggests that a small kernel FWHM (less than two voxels) for fMRI data smoothing reduces the false positive rate and the potential overestimation of cluster size (Liu et al., 2017).”

Liu, Peng, Vince Calhoun, and Zikuan Chen. "Functional overestimation due to spatial smoothing of fMRI data." *Journal of Neuroscience Methods* 291 (2017): 1-12.

Wang, Hongye, et al. "The longitudinal relationship between BOLD signal variability changes and white matter maturation during early childhood." *Neuroimage* 242 (2021): 118448.

Reviewer #2 (Remarks to the Author):

2.1.

Throughout the manuscript, the authors argued that group size increases cooperation. However, it is unclear whether the current design and results support the conclusion. On the design, it is possible that higher cooperation tendency maintains a larger group size, rather than the way around. That is, how many partners a participant had depended on how cooperative they were. On the results, the current study revealed that larger distance-related forgetting resulted in choices based on intrinsic social tendency instead of value associated with group size. Together, it is likely that larger group size one maintains is just a byproduct (not a reason) of one's higher cooperation rates.

Thank you for the comment. We fully agree with there being a risk of group size being, in fact, a byproduct of higher cooperation rates. The claim about group size and cooperation that we make in the paper is associative, not causal. In fact, this is precisely what we aim at showing using our generative modelling approach. Model comparison directly supports this claim, showing that including group size as a predictor in a model which already includes forgetting does not increase explanatory power.

The comment suggests that there is also a possible alternative explanation to the observed pattern: namely, that prosocial tendency drives group size. If this was the case, observing higher cooperation rates in larger group could be the result of a classic selection bias: since only more cooperative participants form those larger groups. This hypothesis leads to 2 clear predictions:

1. The average cooperation should correlate with average group size across participants.
2. Within individuals, group size should NOT affect cooperation rate

The first point relates to between-person variability, while the other, to within-person effects. Both of those are easily testable given our design, and both need to necessarily be true to confirm this alternative explanation.

1. This is indeed true, however the effect size is very small and barely significant (mean correction $r=0.22$, testing $r>0$: $t(81) = 2.08$, $p=0.04$).
2. If we can observe the same participants expressing higher levels of cooperation in larger groups, this is strong evidence that there *exists* a direct or indirect causal mechanism linking increased group size with higher cooperation. This is in fact what we see in our data. To test this directly, we isolated the within-person effect by performing a mixed-effect analysis with group-centered predictors. This procedure removes the between-participant association, and all the remaining variance the model explains relates to the within person effect, see:

Bell, Andrew, Kelvyn Jones, and Malcolm Fairbrother. "Understanding and misunderstanding group mean centering: A commentary on Kelley et al.'s dangerous practice." *Quality & quantity* 52 (2018): 2031-2036.
<https://www.ncbi.nlm.nih.gov/pmc/articles/PMC6096905/>.

Our analysis for this issue indicated a strong, positive within-level effect of group size on cooperation: $\beta=0.093$, $SE=0.015$, $z=6.144$, $p<0.001$.

Together, these two points show that most of the variance in *group size - cooperation* association is attributable to the within-person effect (driven by interaction distance), rejecting the possibility of the effect being driven purely by the between-level effect (*i.e.*, average prosocial tendency driving group size). Please keep in mind that this is consistent with our model-based analysis, which suggests that the within-person group size has an *indirect* effect on cooperation, explained by the direct influence of interaction distance.

This analysis has been added to the Supplementary Materials in the section titled Can the Effect be Explained by Prosocial People Having Larger Groups?

2.2.

The exclusion of 4 participants who defected in most trials should be justified further.

Thank you for pointing this out. We excluded the subjects who always defected, as we deem them to be completely unreceptive to the experimental manipulation. We added the following paragraph to the *Methods* section to better justify our reasoning:

Lines 567-574:

4 participants (all participating in the behavioral-only session) were excluded from further analyses due to defecting on the vast majority (>95%) of trials. While we acknowledge that ‘always defect’ is a viable strategy used by a subset of population in prisoner’s dilemma games²⁰, this a priori selection criterium served to remove participants who were completely unreceptive to the experimental manipulation. It is also very difficult to model subjects who always make the same choice, and would severely affect statistical sampling in the fMRI analysis to have little-to-no contrast in behavior.

2.3.

For the LLM analyses, why effects of GS and ID were estimated in separated models, given that those effects were estimated simultaneously in the computational models?

This model design was chosen because, in the LMM, the collinearity between GS and ID could provide a challenge for reliable parameter estimation. This issue wasn’t a problem for the computational model, since (a) the ID variable was transformed (played the role as the forgetting exponent) and (b) the Bayesian framework of the computational model is more robust to this issue (and easily diagnosable, if fitting problems do arise, by examining posterior Credible Intervals). Good parameter recovery and reliable posterior predictive checks (see: Supplementary Materials, section Recovery of model parameters) confirm that the association between ID and GS was not an issue for the computational model.

2.4.

For the computational modelling, the authors only considered value of cooperation in their valuation functions, so that the value of defection was fixed to 0. However, there are several other factors that may impact participants’ choices: (1) the monetary reward. It is likely that participants made their choices based on the tradeoff between social and monetary reward, but the current models did not capture the tradeoff; (2) the second-order

belief of how their own choices may impact partner's follow-up decisions, see also Hampton et al. (2008) and Hill et al. (2017).

1) This is an important point. Unfortunately, the study design does not allow for disambiguating between monetary reward and value of cooperation (VC). VC can be thought of as the difference between value of cooperation and defection, hence setting the value of VD to 0 is arbitrary. Since the payoff matrix in our experiment was fixed both within and between participants, there's no way of testing the independent effects of reward. The only way of extracting independent effects of reward would be to experimentally vary the payoff matrix, which is a promising idea for a potential follow up.

However, note in Figure S4A that more frequently defecting is more optimal from the score-perspective in our task, yet subjects clearly are playing suboptimally in order to prioritise cooperative interactions. Therefore, the subjects were not purely pursuing the strategy of monetary/point-based payoff.

2) Second-order beliefs, resulting from mentalizing, are indeed crucial in understanding individual strategy. While extremely interesting, we believe that theory of mind, and how depth of mentalizing influences strategy and choice are beyond the scope of this paper. We acknowledge this in the discussion paragraph:

Lines 478-496:

While our model fits the data quite well, we can observe some differences in fit quality across participants (Fig. 3EF). This issue is likely the result of rich individual differences in PD strategies used by human participants²⁰, not all of which could be accurately captured by our model. Specifically, the model would struggle when faced with heuristic strategies which do not rely on reciprocity, such as GRIM or 'always defect'²⁰, or more sophisticated decision rules⁶⁸. We did not find participants who used the GRIM strategy, but a small number did almost exclusively defect as noted in the exclusion criteria, and it is possible more complicated heuristics may be used beyond these. Indeed, certain personality characteristics have been previously associated with specific strategic choices⁶⁹. Additionally, people are

known to use theory of mind to form second-order beliefs and anticipate partners' actions^{70,71}. Individual differences in personality and mentalizing depth provide excellent leads for future studies, which could provide more powerful predictions and understanding. One potential remedy for this in the context of our design is the development of mixture models⁷², which can explicitly model the probability of a participant using one of a set of fixed choice strategies. Mixture models are notoriously difficult to fit due to identifiability and model-selection issues, so caution should be advised⁷³. Our current model fit quality was high, and our main focus was on commonalities, not individual differences, so we did not pursue this avenue further.

2.5.

Please provide relevant references (e.g., theoretical backgrounds, applications, and/or simulations) to justify the application of beta-restricted priors.

The motivation for using beta-restricted priors stems from a few modelling considerations, specific for hierarchical Bayesian modelling.

1. Avoidance of the Neal's Funnel problem

Neal's funnel relates to a situation where sampling the posterior distribution is difficult due to unusual distributional geometry. A classic example is when one parameter value is dependent on a function of another, such as variance of parameter a being dependent on the mean of parameter b (see: https://mc-stan.org/docs/2_18/stan-users-guide/reparameterization-section.html). This situation often occurs in hierarchical models, where individual parameters are drawn from group-level hyper-parameters. A common way of dealing with this problem is using a non-centered parametrization, where all parameters are conventionally drawn from a standard Gaussian, and transformed afterwards. The beta-restricted prior used the same concept, but utilizes Beta(2,2) instead of the standard Gaussian.

2. Parameter Identifiability

The above point raises a question: why not use more conventional standardized Gaussian non-centered priors? For naturally constrained parameters, such as learning rate α and memory parameter k in our models, which are bound between 0 and 1, the answer is quite straightforward: a Beta distribution is the natural sampling distribution of these parameters (since it is also bounded between 0 and 1). For regression parameters, which are technically unbounded, the answer is more subtle. Often, in complex models, parameter values can only be identified on a restricted range:

Guse, Björn, et al. "Assessing parameter identifiability for multiple performance criteria to constrain model parameters." *Hydrological Sciences Journal* 65.7 (2020): 1158-1172.

In addition, allowing parameters to have long tails in complex models expresses the belief that they can potentially have unrealistically high values, which then can lead to estimation problems, such as poor fit or poor recoverability, resulting from giving the model too much flexibility. We restrict regression parameter ranges to a maximum of 5 standard deviations (all predictors were z-scored), which is a range already wider than one should expect (for detailed discussion on parameter ranges see the answer to the next question).

From my (WZ) personal experience with these models, restricting standardised parameters to realistic ranges can improve posterior inference, as well as dramatically speed up the computation. *Parameter recovery analysis* in *Supplementary Materials* shows that the parameter estimates are not biased, and true ranges can be captured accurately, proving the procedure to work well in the context of our model.

We added a brief version of this argument to the *Priors and parametrization* of the *Methods* section, to better motivate this choice:

Lines 768-802:

Priors and parametrization. We use a novel approach referred to as *beta-restricted priors*, where posterior parameter distributions have a constrained value range (*i.e.*, explicitly specified minimal and maximal values) and a continuous distribution. This approach allows us to mitigate common issues with hierarchical Bayesian estimation, such as Neal’s Funnel⁸⁶ (via the use of non-centered fitting) and poor identifiability⁸⁷ (via reducing the parameter ranges to plausible values). For this, we first define priors in ‘raw’ space:

$$\begin{aligned}\alpha_p &\sim \Gamma(1,1) \\ \beta_p &\sim \Gamma(1,1) \\ p_{raw} &\sim B(\alpha_p, \beta_p)\end{aligned}$$

where Γ is the gamma distribution, and p_{raw} is an individual parameter of interest distributed

in accordance with a beta distribution B , with shape parameters α_p and β_p . Then, these *raw* values are transformed to their native space:

$$p = p_{raw} * (UB_p - LB_p) + LB_p$$

where LB_p and UB_p are the lower and upper parameter boundary, respectively. Similarly, we can compute the distributional statistics for the group-level parameter distribution by utilizing α_p and β_p . The posterior group mean is given by:

$$\mu_p = \alpha_p + \beta_p * (UB_p - LB_p) +$$

Intuitively, this approach allows for fine control over parameter ranges, making it easier to avoid issues with parameter identifiability without the need to resort to highly improbable and non-informative uniform priors.

The following parameter ranges were used:

$$V_0^C E (-10, 10)$$

$$V^{RC} E (-10, 10)$$

$$V^{RD} E (-10, 10)$$

$$V^{GS} E (-5, 5)$$

$$V^P E (-5, 5)$$

$$k E (0, 1)$$

$$\alpha E (0, 1)$$

The ranges of V_0^C and V^{RD} were set so that extreme values of cooperation and reciprocity were able to be fitted and recovered (Fig. S7). Ranges of V^{GS} and V^P were set so that they would allow for very strong effects of the linear predictors (up to 5 standard deviations above the mean, as all predictors were z-scored). The ranges of k and α expressed natural parameter bounds. For identifiability purposes, V^{RC} and T parameters were fixed to 1 during model fitting.

2.6.

How the ranges of parameters were set? Please provide justification. What the range of tau (inverse temperature)?

- 1) The ranges of fitted parameters are provided in the method section and are as below:

$$V_0^C \in (-10, 10)$$

$$V^{RC} \in (-10, 10)$$

$$V^{RD} \in (-10, 10)$$

$$V^{GS} \in (-5, 5)$$

$$V^P \in (-5, 5)$$

$$k \in (0, 1)$$

$$\alpha \in (0, 1)$$

- 2) The ranges for V_c , V^{RD} and V^{RC} variables were set to accommodate extreme ranges of cooperation/ reciprocity. This range selection was informed by prior predictive simulations, which showed that by setting one of these parameters to one of the extremes can produce cooperation and reciprocity levels higher/lower to the extremes observed in the actual data, and that these values can be recovered by fitting the synthetic data. The ranges for V^{GS} and V^P were set so that to allow for very strong effects of the linear predictors (up to 5 standard deviations above the mean,

as all predictors were z-scored). Note that effect sizes above 3 standard deviations above the mean Z-scoring predictors have multiple purposes:

- a) Aids model fitting (see: <https://mc-stan.org/docs/stan-users-guide/efficiency-tuning.html#standardizing-predictors>)
- b) Allows for common, interpretable priors
- c) Facilitates posterior parameter interpretability

Parameters α and k are naturally bounded between 0 and 1.

To make this reasoning explicit we added the following paragraph:

Lines 797-802:

The ranges of V_0^C and V^{RD} were set so that extreme values of cooperation and reciprocity were able to be fitted and recovered (Fig. S7). Ranges of V^{GS} and V^P were set so that they would allow for very strong effects of the linear predictors (up to 5 standard deviations above the mean, as all predictors were z-scored). The ranges of k and a expressed natural parameter bounds. For identifiability purposes, V^{RC} and T parameters were fixed to 1 during model fitting.

- 3) Inverse temperature (τ) was fixed to 1 for identifiability. This is because, in our model, the values of cooperation and defection are not independent (value of defection is fixed to 0), reducing the choice function to a logistic model with a single parameter. This is the consequence of having a single latent variable representing the internal value of cooperation. An alternative approach would be to model values of defecting and cooperating as independent variables, driven by different inputs. This is however, in our eyes, a much less parsimonious assumption, and would require strong theoretical justification.

We realised that the fact of fixing τ to 1 was not explicitly mentioned in the model specification, hence we fix this ambiguity in the main text:

Lines 747-748

Since the value of defection is fixed, this formulation is equivalent to using the logistic function, rendering the inverse temperature unidentifiable. Hence, τ was fixed to 1.

2.7.

VRC were fixed to 1, please provide justification. The identifiability issue may be due to the limited number of data or the problem of models. The authors may provide more detailed descriptions about the issue.

This fixed value is due to the fact that this parameter proved to be impossible to recover, due to extremely high negative collinearity between the two variables, VRC and VRD. These variables are impossible to disentangle because their predictor variables, probability of previous cooperation (PC) and probability of previous defection (PD) are mirror images of

each other. This issue is a consequence of model structure, namely the fact PC and PD need to sum to 1. Therefore, if $PC=x$, PD has to be $1-x$ (or *vice versa*). Hence there is only one degree of freedom and the variables are necessarily collinear, hence only one of them can be reliably estimated. For current model recovery performance see *Recovery of model parameters* section of *Supplementary Materials* and figure S7.

2.8.

For the GLM3, would value of forgiveness and value of betrayal be highly correlated with each other?

There's not much of a reason to expect this, since betrayal and forgiveness are exclusive events. Value of betrayal can only happen after previous cooperation by a partner, while value of forgiveness only after previous non-cooperation by a partner. So the non-zero values are exclusive subsets (when the value of forgiveness is non-zero, value of betrayal has to be 0, and *vice versa*). To confirm this, we performed a correlation analysis per subject, which indicated an overall negative, but variable pattern of correlation ($M(\text{correlation}) = -0.29$, $SD = 0.31$) between the two parameter values, with no clear pattern.

2.9.

Many ROIs were constructed, but the authors did not provided relevant hypotheses on why specific ROIs were chosen.

Thank you for making us realise that we did not express clearly why we have chosen particular ROIs. We expanded the ROI Methods section to make this clear:

Lines 893-931:

Value of cooperation. The VMPFC ROI was defined as 10 mm^3 sphere, centered at the peak activity for the keyword 'value' (peak MNI coordinates: $x=0$, $y=40$, $z=-8$; 407 studies included in the database). Choice of VMPFC was driven by the *a priori* expectation of the region being involved in subjective value processing³¹ (including social contexts⁹⁰).

Forgetting. This set of ROIs was chosen in order to test the connectivity between memory representation (precuneus) and value representations, and included: precuneus, FFG, NAcc and dACC.

The precuneus ROI was defined as 10 mm^3 sphere, centered at the peak activity for the keyword 'memory' (peak MNI coordinates: $x=-8$, $y=-66$, $z=28$; 2744 studies included in the database). The choice of precuneus was motivated by both: an *a priori* expectation of the region playing a crucial role in social memory⁴⁰ as well as the results of earlier model-based analyses, relating its activity to forgetting.

The FFG ROI was defined as two 6 mm³ spheres, centered at the peak activity for the keyword ‘*face recognition*’ (peak MNI left: x=-42, y=-52, z=-20; peak MNI right: x=41, y=-52, z=-20; 79 studies included in the database). The choice of FFG was motivated by both: an *a priori* expectation of the region being involved in face recognition and memory^{56,57}, as well as the results of earlier model-based analyses, relating its activity to forgetting.

The NAcc ROI was defined as two 5 mm³ spheres, centered at the peak activity for the keyword ‘*value*’ (peak MNI left: x=-6, y=10, z=-5; peak MNI right: x=6, y=-10, z=-5; 407 studies included in the database). The choice of NAcc was motivated by both: an *a priori* expectation of the region being involved in reward processing⁵¹, as well as the results of earlier model-based analyses, relating its activity to the value of cooperation.

The dACC ROI was defined as a 10mm³ sphere centered at the peak activity for the keyword ‘*control*’ (peak MNI: x=4, y=18, z=42; 3796 studies included in the database). The choice dACC was motivated by both: an *a priori* expectation of the region being involved in integrating value information across timescales^{38,39}, as well as the results of earlier model-based analyses, relating its activity to the value of reciprocity.

Congruence. For the congruence analysis, we re-used the precuneus and dACC ROIs defined above. Additionally, we built 2 ROIs for the left and right DLPFC, defined as 5 mm³ spheres, centered at [-40, 40 24] and [40, 40 24], respectively. Similarly, to the earlier definition of dACC, we used *Neurosynth* peak coordinates related to the term ‘*control*’, within clusters which anatomically included the DLPFC region. We chose those regions for both theoretical and practical reasons (*i.e.*, significant whole-brain activity related to key functional roles in prior GLM models). The DLPFC and dACC have been implicated in representing value-based information across different timescales^{38,39}, while the precuneus is a central hub of the mentalizing network, suggested to play a role in social working memory^{34,90}.

We also added additional context in the main text:

Lines 289-291:

Since *VtC* represents subjective value, we also expected to see signals related to it within the ventromedial prefrontal cortex (VMPFC), area heavily implicated in processing subjective value signals³¹

Lines 345-349:

We additionally tested a set of independently defined ROIs from candidate regions which we *a priori* hypothesized to potentially play a role, including the dACC, DLPFC and the precuneus. The DLPFC and dACC have been implicated in representing value-based information across different timescales^{39,40}. The precuneus is a central hub of the mentalizing network, suggested to play a role in social working memory³⁸.

2.10.

Some typos in the manuscript:

Abstract: ‘...one’s individual prosocial preference in a core pillar’ should be ‘is a core pillar’

Discussion: ‘...and plays a major role in brain state transitions’ should be ‘in brain state transitions’.

Thank you for pointing these out, the typos have been corrected.

Reviewer #3 (Remarks to the Author):

This article describes a novel paradigm described as assessing the role of time and forgetting in cooperation in increasingly large groups. The overall conceptualization and implementation of the study is novel and likely to be of wide general interest.

I would also like to say that the main text of the article is very well written... the authors should be commended on crafting a manuscript that was such a pleasure to read.

We are extremely happy to hear that reading our manuscript was a pleasant experience. Thank you for the encouraging comment!

I do have a few concerns about the methods and the conclusions that can be drawn from the results, as well as some thoughts about the framing of the article, which I describe in more detail below:

3.1.

Introduction:

- Overly specific lists of regions and their function on p6--seems to suggest that, e.g., DLPFC plays some unique and specific role in betrayal. Does it?

The specific list of regions relates to our functional results, where ‘*betrayal*’ is defined specifically as defecting after a given partner cooperated during the previous interaction with the same partner. We claim that DLPFC plays a role in betrayal, as we observe heightened levels of DLPFC activity during reciprocity-breaking defections. Please note that, in this sense, ‘*betrayal*’ doesn’t necessarily convey intentionality; it can just as well be a result of forgetting, or misremembering the partner’s previous action. In addition, the DLPFC is implicated to overriding predominant tendency (Ref 63). This definition of betrayal is used consistently throughout the manuscript.

We have bolstered our discussion of prior literature relating DLPFC to betrayal-related behaviors in the new section:

Lines 461-467:

“Previous literature has also linked DLPFC to functions that are conceptually related to betrayal, such as cognitive control⁶⁰ and modulation of social versus anti-social economic behaviors⁶⁶. For further example, left DLPFC has been linked to negative emotional responses to images⁶¹, as well as children making incorrect choices to adult instructions⁶⁰. Specifically in the Prisoner’s Dilemma, the DLPFC has been implicated in the level of guilt felt after unreciprocated cooperation⁶⁷.

3.2

Methods/Results

- On p12 of the Discussion the authors state: "our model predicts that the memory trace of the current partner’s previous choice decays with time." Unless there was a design feature that I missed, time is confounded with the number of prior interactions in this paradigm. In theory, different memory processes would be associated with forgetting depending on the cause. Can the authors distinguish whether the cause of forgetting was the passage of time (WM) versus the number of intervening interactions (interference). I agree with the authors' statement in the Limitations that a lack of any explicit tests of memory phenomenon are a serious limitation of a study that makes strong claims, starting in the title, that the observed effect reflects specific memory mechanisms.

Yes, the comment is correct in assuming that time between interaction with the current partner is confounded with the number of intervening trials (interaction distance). Since both are almost perfectly correlated, it would be impossible to disentangle their relationship. Future work could provide a more systematically tuned inter-trial interval (ITI) to untangle this relationship, but it is not possible to untangle them in our current paradigm.

For completeness, we also note that the passage of time (as approximated by trial number across a session) by itself did not predict changes in the level of cooperation (Figure 1D). Hence, we argue that we can also exclude the possibility of session time being the causal factor driving observed changes in behaviour.

To verify this, we performed two simple analyses, adding a short section to the *Supplementary Materials*:

Effects of Time

Potential concern can be raised as to whether observed changes in cooperation and reciprocity can be attributable to drift over time within session. To test this, we correlated trial number with both outcome variables per participant. For cooperation, we found an average correlation of 0.026 95% CI [-0.008, 0.061], and for reciprocity, an average

correlation of 0.015 95% CI [-0.01, 0.036]. We conclude that there's no evidence to suggest this might be the case.

3.3

- *The Methods states that the avatar faces were taken from Ekman face set neutral faces, but the images in Fig 1 are from the NimStim set. Which is correct?? In either case, I wonder why that set was used. Usually that set is selected for its emotional faces. Were there other conditions in which the emotional faces were used that are not being reported?*

We apologize for this inconsistency, the faces were indeed taken from the neutral face set of NimStim dataset, and not from Ekman's face set. This error has now been corrected in the main text (lines 552-553). We only used neutral faces, as we did not aim to test any effects of emotions in this particular study. We added the full stimulus set as a supplementary figure:

Figure S1. All 30 neutral faces used in the experiment. Each participant interacted with the same set of partner faces, introduced in random order. All stimuli were chosen from the neutral set of faces from the NimStim set of facial expressions (Tottenham et al., 2009)

3.4

- *I note that the study is not preregistered and wonder why, and whether the authors can provide additional information about which hypotheses and methods were pre-specified versus ad hoc. (For example, I found it surprising that there was no evidence of hippocampus playing a role in forgetting or episodic memory during the task, and wondered if that was really what was predicted.)*

The main study was first implemented in 2018 when pre-registration was not yet the global norm in neuroeconomics research. We apologise that it did not occur to us at the time and will pre-register future work.

Thank you for pointing out that some of our hypotheses and choices were not motivated clearly enough in the main text. We have now addressed this issue in several parts of the paper (see below).

In GLM 1, while testing effects of value of cooperation, our *a priori* prediction was that VMPFC would be involved in encoding value, which was backed by rich prior literature (for a review see: reference 31), as well as meta-analytic search done in Neurosynth.

In GLM2, while testing the effects of memory, we expected the involvement of both the precuneus and FFG, but not necessarily the hippocampus. While the hippocampus is unambiguously an important memory-related hub, we found little evidence for its role in similar social working memory tasks. Many prior studies on social memory either focus on different areas (e.g.

<https://www.pnas.org/doi/full/10.1073/pnas.1121077109>)

or don't find much evidence for direct hippocampal involvement (e.g.

<https://www.ncbi.nlm.nih.gov/pmc/articles/PMC8412184/>). In fact, we found considerably stronger a priori evidence for the precuneus being central for social memory processing within the fMRI literature (Refs: 34-37, 40). These expectations are expressed in the following lines:

Lines 317-320:

We hypothesized the precuneus to be the central node of this information transfer due to its strategic location as a densely connected functional hub³⁴, as well as its relation to episodic memory retrieval³⁵⁻³⁷, especially in social contexts³⁸.

Line 423:

FFG plays a direct role in face recognition⁵⁶ and memory⁵⁷.

In GLM4, we addressed how connectivity between key hubs involved in memory (FFG, Precuneus) and value processing (NAcc) relates to forgetting. Here, the choice of ROI was driven as much by prior expectations (see: justification for FFG and precuneus in GLM2) as well as the whole-brain activation patterns observed in earlier models. More specifically, observing NAcc in the value-condition, and dACC in the reciprocity condition made those areas our targets for these analyses. Two caveats are to be made here: 1) please note that all ROIs were defined independently from the functional activation patterns: based on meta-analytic coordinates from Neurosynth. 2) choice of dACC instead of AI (which was also active in the reciprocity contrast) was motivated by its central role in information integration across timescales:

Lines 346-347:

The DLPFC and dACC have been implicated in representing value-based information across different timescales^{39,40}

In GLM 5, we tested regions sensitive to the congruence effect. We then tested few a priori defined ROIs, as mentioned in the main text:

Lines 344-348:

We additionally tested a set of independently defined ROIs from candidate regions which we *a priori* hypothesized to potentially play a role, including the dACC, DLPFC and the precuneus. The DLPFC and dACC have been implicated in representing value-based information across different timescales^{39,40}. The precuneus is a central hub of the mentalizing network, suggested to play a role in social working memory³⁸.

To make these motivations clear, we expanded the ROI *Methods* section to include unambiguous motivations for using them.

Lines 893-931:

Value of cooperation. The VMPFC ROI was defined as 10 mm³ sphere, centered at the peak activity for the keyword ‘*value*’ (peak MNI coordinates: x=0, y=40, z=-8; 407 studies included in the database). The choice of VMPFC was driven by the *a priori* expectation of the region being involved in subjective value processing³¹ (including social contexts⁹¹).

Forgetting. This set of ROIs was chosen in order to test the connectivity between memory representation (precuneus) and value representations, and included: precuneus, FFG, NAcc and dACC.

The precuneus ROI was defined as 10 mm³ sphere, centered at the peak activity for the keyword ‘*memory*’ (peak MNI coordinates: x=-8, y=-66, z=28; 2744 studies included in the database). The choice of precuneus was motivated by both: an *a priori* expectation of the region playing a crucial role in social memory³⁸ as well as the results of earlier model-based analyses, relating its activity to forgetting.

The FFG ROI was defined as two 6 mm³ spheres, centered at the peak activity for the keyword ‘*face recognition*’ (peak MNI left: x=-42, y=-52, z=-20; peak MNI right: x=41, y=-52, z=-20; 79 studies included in the database). The choice of FFG was motivated by both: an *a priori* expectation of the region being involved in face recognition and memory^{56,57}, as well as the results of earlier model-based analyses, relating its activity to forgetting.

The NAcc ROI was defined as two 5 mm³ spheres, centered at the peak activity for the keyword ‘*value*’ (peak MNI left: x=-6, y=10, z=-5; peak MNI right: x=6, y=-10, z=-5; 407 studies included in the database). The choice of NAcc was motivated by both: an *a priori* expectation of the region being involved in reward processing⁵¹, as well as the results of earlier model-based analyses, relating its activity to the value of cooperation. The dACC ROI was defined as a 10mm³ sphere centered at the peak activity for the keyword ‘*control*’ (peak MNI: x=4, y=18, z=42; 3796 studies included in the database). The choice dACC was motivated by both: an *a priori* expectation of the region being involved in integrating value information across timescales^{39,40}, as well as the results of earlier model-based analyses, relating its activity to the value of reciprocity.

Congruence. For the congruence analysis, we re-used the precuneus and dACC ROIs defined above. Additionally, we built 2 ROIs for the left and right DLPFC, defined as 5 mm³ spheres, centered at [-40, 40 24] and [40, 40 24], respectively. Similarly, to the earlier definition of dACC, we used *Neurosynth* peak coordinates related to the term ‘*control*’, within clusters which anatomically included the DLPFC region. We chose those regions for both theoretical and practical reasons (*i.e.*, significant whole-brain activity related to key functional roles in prior GLM models). The DLPFC and dACC have been associated with processing value-based information across different timescales^{39,40}, while the precuneus is a central hub of the mentalizing network, suggested to play a role in social working memory^{34,92}.

3.5

- Also, these are somewhat puzzling faces to pick. In either case (Ekman or NIMSTIM) they are of Americans and it seems like this study was run in Japan. Some explanation for this choice of stimuli is required, and at least a brief discussion of the limitations it creates. One big one is of course that Americans (who can often be recognized as such regardless of race from their facial behavior) are 'outgroup' members for people in Japan--at least, they would be described as outgroups by default in many studies. And in general outgroup members elicit less trust and thus cooperation, particularly in more collectivist societies like Japan. So this study design seems like it very likely biased behavior typical to baseline. Do the authors have any evidence to confirm that the choice of these avatar faces did not change participant behavior relative to faces of Japanese students or non-outgroup faces (cartoon characters, abstract images)

First, we confirm that the study was indeed conducted in Japan on Japanese subjects. While we do not have evidence that participants would act differently given different stimuli, we don't expect their behavior to be affected by the ‘outgroup’ perception of the avatars. If this was the case, we would expect to observe relatively low cooperation and reciprocity and values, which was not the case (see Figure 2). While we cannot conclude whether participants would act differently given a different set of avatars, we can say that the current set resulted in general high levels of trust and reciprocity. This suggests that, in the least, the

more foreign-looking avatars (e.g. non-Asian) did not inhibit cooperative behavior. Whether a more familiar set of avatars would bias behavior even more strongly towards cooperation is an interesting question, but not within the scope of our study. Taking this in mind, please note the faces used (as shown in the new Supplementary Figure) are ethnically diverse and of male and females, and all have neutral expressions.

To test potential effect of different faces, we performed an additional set of analyses, which showed that:

- Cooperation levels based on partner's face varied from 0.459 (face number 5) to 0.677 (face number 1) with a mean of 0.573 and SD of 0.058
- Reciprocity levels based on partner's face varied from 0.625 (face number 22) to 0.757 (face number 4) with a mean of 0.699 and SD of 0.032
- The correlation between these two measures (how likely one is to cooperate with a given partner and how likely he is to reciprocate) were correlated at $r = 0.333$, $t(28) = 1.872$, $p=0.07$.
- Between-person ANOVA on Cooperation rate as predicted by Gender and Race, revealed no significant effects of Gender $F(1,26)=1.045$ $p=0.366$, nor Race $F(2,26)=0.231$ $p=0.635$
- Between-person ANOVA on Reciprocity rate as predicted by Gender and Race, revealed no significant effects of Gender $F(1,26)=0.266$ $p=0.590$, nor Race $F(2,26)=0.539$ $p=0.610$.

Together, this suggests that, within the scope of the faces that were used for the current experiment, the gender nor race of the partner did not influence behavior to a significant degree. We added these analyses to the *Supplementary Materials* (section: *Effects of Partners' Faces*).

3.6.

- *Why was age not recorded for 1/3 of participants?*

We regrettably did not enforce the complete pre-experiment survey forms to be filled out in their entirety, and thus several subjects did not feel comfortable providing their age for some reason and omitted it. However, the experiment was conducted in the college setting (in the campus of Osaka University) and visually all subjects were approximately college-aged young adults.

3.7.

- *I'm confused by the line: "The partner was given the same stay/switch decision and was more likely to switch if the subject was more defective recently." I thought the partners were "prescheduled computer programs"? How exactly was this decision programmed in? It seems like a lot of the study is riding on the precise "behavior" and "decisions" of the computer that is not entirely clearly specified.*

We have uploaded the full Matlab code that guides the partner. Additionally see the methods section: *"Algorithm of Social Partner"*. As a simple summary: "we programmed the "attitude" of the computer agent to change dynamically. This dynamic nature came from two sources in the agents' algorithm: one from the learning process provided by a reinforcement learning-

like choice algorithm and another from cyclical alternations between the cooperative and non-cooperative strategies.”

Most relevant partner algorithm code snippets:

```
%Decision variable
Mood = Network.P(PlayerNum).Mood(TrialNum); %PlayerNum = Partner ID
DV_CD = Network.P(PlayerNum).V_SOC + B_CD + Mood;

%Softmax normalization for action probability
P_CD = 1 / (1 + exp(-DV_CD/Sigma))

%Stochastic Process
D_CD = binornd(1,P_CD);

if D_CD == 1
    DEC = 'Cooperate';
elseif D_CD == 0
    DEC = 'Defect';
end
...
%Value calculation from the past history of interaction with the current
partner
% Individual Learning
Network.P(PlayerNum).V_SOC = Network.P(PlayerNum).V_SOC +
ML_SOC*(Network.P(PlayerNum).Self(end) - Network.P(PlayerNum).V_SOC);

%Memory or learning rate (weights for the past experiences)
ML_SOC = 0.3;

Sigma = 1.0;%Stochasticity of choice
B_CD = 0.0; %C-D Bias

MoodGain = 3;
Network.P(N).Mood = MoodGain*sin(-(10+N)*pi:0.3:(10-N)*pi);
```

3.8.

- I'm less sure that this study is capturing actual group dynamics if people only ever interact with computers. It seems like human behavior might be shaped by a context like this in a way that would meaningfully change prosocial responding, which can't be tested when each human responder is only interacting with computers, especially if all "partners" were programmed with same algorithm. A key feature of real groups, and the decision to cooperate in sequential prisoner's dilemmas, is variation.

Importantly, the cyclic mood parameter which had a period of ~20 trials was not synced between partners, thus different partners had different initial baseline tendencies to cooperate or defect that slowly drifted, while the partners also had an RL learning equation to interact closely with recent subject choices on top of this. Post-experiment interviews

suggested these designs were sufficient to convince subjects that they were playing with humans. The experimental team also tested the task extensively before running the experiment and their view was the algorithm felt dynamic but strategic like a real human, rather than too simple or too random.

3.9

- I'm also not sure if I'd call this a "group" in any meaningful way. As far as I understand it, each person is only aware of their own ongoing interaction with each avatar but has no idea if they are interacting with each other or not, or have any other kind of collective identity, which would be intrinsic to it being a "group."

Yes, the comment is correct in assuming that group structure is different and private for each participant. The design attempts to control for factors such as reputation, which could affect behavior if the interactions weren't private. Also, groups that are common among sets of participants would make '*breaking ties*' mechanic difficult to implement. We think this arrangement is functionally equivalent to a group where there's little prior knowledge about other members, and allows for better control of potential confounders. Additionally, post-experiment interviews suggested our task design was sufficient to convince subjects that they were playing with humans. Furthermore, the subjects arrived as a group and met each other before the game started as discussed in the methods section. During the experiment, the avatars of all current group members were displayed on the screen (as shown in Figure 1), so that the subjects would feel that they were a part of a small group.

3.10

- I had some concerns upon reading the line: "The subjects overall seemed exhausted but appeared to enjoy the experience in the task by the end." How many trials were in the task that rendered participants 'exhausted'? I searched for the length of the study or number of trials and could not locate that information. Is there a concern that this exhaustion is part of the 'forgetting' that might be independent of the specific proposed memory-based mechanisms?

The task was a standard length for a neuroeconomics study. The sessions lasted for 180 trials, and the average completion time was around 45 minutes. We added this information to Lines 582-583.

In hindsight, we should not have written that the subjects were exhausted by the end of the task. This was originally meant to convey that, from our observations, the subjects put a lot of effort into the task and thus were engaged with the dynamic nature of social interactions in the context of frequent changes of the social connections. We now unfortunately realize this reads as if they were getting fatigued, bored, overwhelmed etc., which is not the case, and we have metrics supporting the opposite actually. We have removed this part of the text. We now show that RTs are stable across the session time and thus we do not think the subjects are getting detrimentally fatigued and apologize for the mis-writing.

Still, if exhaustion was present, we would likely expect responses to become progressively slower, which was not the case. Below we plot average RT as a function of trial (gray ribbon around the estimate represents +/- 1 SD):

This plot has been added to the *Supplementary Materials* as Figure S8.

3.11

- As far as the neural substrates go, can the authors explain why the DLPFC would have a special role in responding to betrayal given this region's involvement in "forgiveness" as well? Hard to say for sure from the images, but the activation patterns appear to be overlapping. Could this region be doing something more general related to reciprocity? The functional connectivity analysis seems potentially consistent with that. (A contrast between responses to forgiveness versus betrayal would seem to be helpful here)

Yes, our belief is that DLPFC's function, in this context, is more general than simply biasing choices towards betrayal or forgiveness. Rather, we think it's related to integration of information across different timescales, consistent with prior literature (Refs 38, 39)

This information then can be used in the decision process, leading to informed departure from the 'default' action (Ref 63).

For interpreting the connection between the DLPFC and betrayal, we've focused on both (1) multi-timescale computational role of the DLPFC and (2) the connection in prior literature to the DLPFC in terms of betrayal relevant concepts such as emotions and guilt, antisocial behavior, and executive control.

For (1), we've referenced and cited related literature extensively so will not expand unless further asked, see *Crucial roles of DLPFC and dACC in information integration across timescales* section of the *Discussion*; lines 433-467)

For (2), the connection to DLPFC with betrayal-related neural signals we've expanded the references and discussion with this new content:

Lines 461-467:

Previous literature has also linked the DLPFC to functions that are conceptually related to betrayal, such as cognitive control⁶⁰ and modulation of social versus anti-social economic behaviors⁶⁶. For further example, left DLPFC has been linked to negative emotional responses to images⁶¹, as well as children making incorrect choices to adult instructions⁶⁰. Specifically in the Prisoner's Dilemma, the DLPFC has been implicated in the level of guilt felt after unreciprocated cooperation⁶⁷.